# A deep learning approach reveals unexplored landscape of viral expression in cancer

Abdurrahman Elbasir[1], Ying Ye[1], Daniel E. Schäffer[1,2], Xue Hao[1], Jayamanna Wickramasinghe[1], Konstantinos Tsingas[1,3], Paul M. Lieberman[1], Qi Long[3], Quaid Morris[4], Rugang Zhang[1], Alejandro A. Schäffer[5] & Noam Auslander[1] ✉

About 15% of human cancer cases are attributed to viral infections. To date, virus expression in tumor tissues has been mostly studied by aligning tumor RNA sequencing reads to databases of known viruses. To allow identification of divergent viruses and rapid characterization of the tumor virome, we develop viRNAtrap, an alignment-free pipeline to identify viral reads and assemble viral contigs. We utilize viRNAtrap, which is based on a deep learning model trained to discriminate viral RNAseq reads, to explore viral expression in cancers and apply it to 14 cancer types from The Cancer Genome Atlas (TCGA). Using viRNAtrap, we uncover expression of unexpected and divergent viruses that have not previously been implicated in cancer and disclose human endogenous viruses whose expression is associated with poor overall survival. The viRNAtrap pipeline provides a way forward to study viral infections associated with different clinical conditions.

Viral infections have a causal role in ~15% of all cancer cases worldwide[1]. Viruses linked to cancer are generally divided into direct carcinogens, which drive an oncogenic transformation through viral oncogene expression, and indirect carcinogens, which may lead to cancer through mutagenesis associated with infection and inflammation. To date, seven viruses have been classified as direct carcinogenic agents in humans[2]. Among these, the high-risk subtypes of human papillomavirus (HPV) are the causative agent of ~5% of human cancers. Chronic hepatitis B virus (HBV) or hepatitis C virus (HCV) infections are associated with most hepatocellular carcinoma cases. More recently, advances in sequencing technologies have contributed to a better appreciation of the high burden of viral infections in cancer, exemplified by Kaposi's sarcoma herpesvirus and the Merkel cell polyomavirus, which were discovered based on nucleic acid subtraction to cause Kaposi's sarcoma and Merkel cell carcinoma, respectively[2]. The discovery of oncogenic viruses, starting with the Rous sarcoma virus[3], has been critical for understanding mechanisms driving cancer evolution and for improving cancer prevention and intervention strategies. However, the burden of viral infections in cancer is thought to remain underappreciated by much of the cancer research community[4].

Since the advent of next-generation sequencing, new viral strains are typically identified from large-scale DNA or RNA sequencing data based on sequence similarity to known viruses. The Cancer Genome Atlas (TCGA) has become a principal resource for the identification of viral sequences in cancer tissues. Several studies screened TCGA DNA sequencing data to characterize known viruses in cancers[5], and analyze host integration sites for viruses such as HBV that integrate into the human genome[6]. Other studies used RNA sequencing to screen for known viruses in the human transcriptome[7–10], and to discover novel viral isolates[10]. Most recently, a few studies combined DNA and RNA sequencing to quantify the presence of known cancer-associated viruses in human cancers[11,12]. However, the set of sequenced viral clades and the set of viral clades known to infect humans are both incomplete. Viruses and cancers have rapidly evolving genomes, and a

[1]The Wistar Institute, Philadelphia, PA 19104, USA. [2]Computational Biology Department, Carnegie Mellon University, Pittsburgh, PA 15213, USA. [3]University of Pennsylvania, Philadelphia, PA, USA. [4]Computational and Systems Biology, Sloan Kettering Institute, New York City, NY 10065, USA. [5]Cancer Data Science Laboratory (CDSL), National Cancer Institute, National Institutes of Health, Bethesda, MD 20892, USA. ✉e-mail: nauslander@wistar.org

new cancer-associated virus may have little sequence similarity to known viruses isolated outside of the tumor microenvironment. This issue is exacerbated when analyzing short reads, which are typical of RNA sequencing technologies. Therefore, the discovery of new and divergent cancer viruses remains highly challenging with existing strategies[13]. For the detection of bacterial viruses from metagenomic DNA sequencing, several machines and deep learning techniques have been recently developed. These methods overcome some of the limitations associated with homology-based approaches and rapidly identify viral reads including novel and divergent viruses[14–18]. More recently, methods have been developed to identify viruses that have the potential to cause human infections[19,20]. These recently developed methods suggest that deep learning methods to detect viral reads from RNA sequencing have the potential to uncover novel and divergent viruses in human tissues.

Here, we develop a framework, named viRNAtrap, that employs a deep learning model to accurately distinguish viral reads from RNA sequencing, and utilizes the model scores to assemble viral contigs. We apply viRNAtrap to 14 cancer types from TCGA (selected based on potential viral relevance to oncogenesis), to perform exploratory data analysis and characterize the landscape of viral infections in the human cancer transcriptome. We demonstrate the ability of viRNAtrap to identify different types of viruses that are expressed in tumors by constructing three viral databases and comparing viRNAtrap findings to sequences in those databases. We first evaluate known cancer-associated viruses that are expressed in different tumor types. Then, we curate a database of potentially functional human endogenous retroviruses (HERVs) and analyze expression patterns of different HERVs across human cancers to find that HERV expression is associated with poor survival rates. Finally, we employ viRNAtrap to identify divergent viruses that are expressed in tumor tissues. Notably, we identify a *Redondoviridae* member that is expressed in head and neck carcinomas, a *Siphoviridae* member that is expressed in 10% of high-grade serous ovarian cancers, and a *Betairidovirinae* member that is expressed in more than 25% of endometrial cancer samples. In summary, we present the first deep learning-based method to identify viruses from human RNA sequencing and demonstrate its ability to rapidly characterize viruses that are expressed in tumors and uncover viral instances that have not been previously found in these samples using alignment-based methods. viRNAtrap can be applied to identify new viruses that are expressed in a variety of other malignancies, introducing new avenues to study viral diseases.

## Results
### The viRNAtrap framework
To identify viruses in the human transcriptome, we first trained a neural network to distinguish viral reads based on short sequences. We collected positive (viral) and negative (human) transcripts that were segmented into 48 bp fragments and divided into training and test sets (Fig. 1a, Methods). We used different metrics to evaluate the ability of the model to identify viral sequences based on short segments. The model yielded test-set performance: area under the receiver operating characteristic curve (AUROC) of 0.81, area under the precision-recall curve (AUPRC) of 0.82 (Fig. 1b), the accuracy of 0.71, recall of 0.83, the precision of 0.67 and F1-score of 0.74 (Fig. 1c). We compared the performance of this model to previous models trained to identify viruses, namely DeepViFi[16], DeepVirFinder[15], ViraMiner[21], as well as a method called "off-the-shelf Seq2Seq" compared through DeepViFi[16], that does not use much domain-specific knowledge about viruses (Methods). Importantly, our model outperformed other methods in all measures, except for precision, for which DeepVirFinder outperformed all other methods (Fig. 1b, c). However, precision is less critical for this framework because alignment steps are used to further filter out negatives. Importantly, DeepViFi[16], DeepVirFinder[15], and ViraMiner[21] were previously not trained or evaluated for RNA

sequencing or 48 bp reads, which is likely the reason that these methods are less appropriate in that context without specific optimization (see Methods). Examining the average model performance across segments from different human viruses, we find that human single-stranded DNA viruses from taxon *Monodnaviria* were assigned with high confidence, whereas, for RNA viruses, we observed more variation in model confidence. For example, the model confidently predicted the viral origin of sequences from Ebola and influenza viruses but assigned borderline scores to sequences from several *Phenuiviridae* members such as *Dabie bandavirus* (Fig. 1d and Supplementary Data 1).

Based on the trained neural network, we built a computational framework (Fig. 1a, Methods) to identify viral contigs from tumor RNAseq and applied the framework to 7272 samples from 14 cancer types in The Cancer Genome Atlas (TCGA)[22], from which 6717 were tumor samples and 555 were non-cancer samples matched to a cancer sample from the same individual (Supplementary Data 2). In pre-processing, we extracted reads that were not aligned to the human genome (hg19) or to the phiX phage[23] that was identified as a frequent contaminant. The computational framework, named viRNAtrap, was then applied to unaligned RNA reads (to reduce the running time of viRNAtrap), to detect viral reads and assemble predicted viral contigs. Finally, in post-processing analysis, we used blastn[24] to compare the assembled viral contigs to three curated viral databases. We identified viral contigs originating from reference viruses that are expected in cancer tissues, human endogenous viruses, and candidate novel or more divergent viruses, which are expressed in different cancer types

### Identifying reference tumor viruses
We first characterized the presence of known cancer-associated human viruses in different tumor types. High-risk human *Alphapapillomavirus* strains (HR-αHPVs) were most frequently detected; the type observed in the majority of TCGA samples is HPV16. This is expected because HR-αHPVs, such as HPV16 and HPV18, underlie ~5% of cancer cases worldwide[25] while low-risk human *Alphapapillomavirus* (LR-αHPV) strains, such as HPV54 and HPV201, are mostly associated with the development of genital warts but not cancer[26]. We found at least one HR-αHPV in 288 CESC samples (286 squamous cell carcinoma samples and two non-cancer samples). We found 61 HNSC samples, and a total of 14 samples across other cancer types, that contain a contig from at least one HR-αHPV (Fig. 2a). LR-αHPVs were identified in a small set of samples mostly from matched non-cancer tissues, including cervix and head and neck (Fig. 2a and Supplementary Data 2, 3).

Hepatitis B virus (HBV) is the second most frequently detected virus across TCGA samples. HBV infections and Hepatitis C virus (HCV) infections are two primary causes of liver cancer and may co-occur in a patient[11]. We found HBV expression in 85 LIHC tumor samples and seven non-cancer samples, and HCV in 13 LIHC tumor samples. HBV was also found in a few tumor samples and matched non-cancer samples from other cancer types (Fig. 2a). By comparing the samples predicted as virus-positive by viRNAtrap to the samples annotated as virus-positive in the TCGA clinical annotations, we found that the true positive rates of viRNAtrap were above 95% for HR-αHPVs (in CESC and HNSC), and for HCV and HBV in LIHC, supporting that viRNAtrap correctly identifies samples expressing known cancer viruses (Supplementary Fig. 1). In addition, viRNAtrap found adeno-associated virus 2 (AAV2) in eight LIHC samples, six from tumors and two from non-cancer samples. AAV2 is a small DNA virus that has the potential to integrate into human genes and contribute to oncogenesis, although the current evidence is insufficient for AAV2 to be included in the consensus list of oncogenic viruses[27,28]. A recent study that addressed discrepancies in AAV2 expression across TCGA samples found at least one AAV2 read in 11 LIHC samples[27]. However, in three of these samples only one AAV2 read was found, which is difficult to detect with the

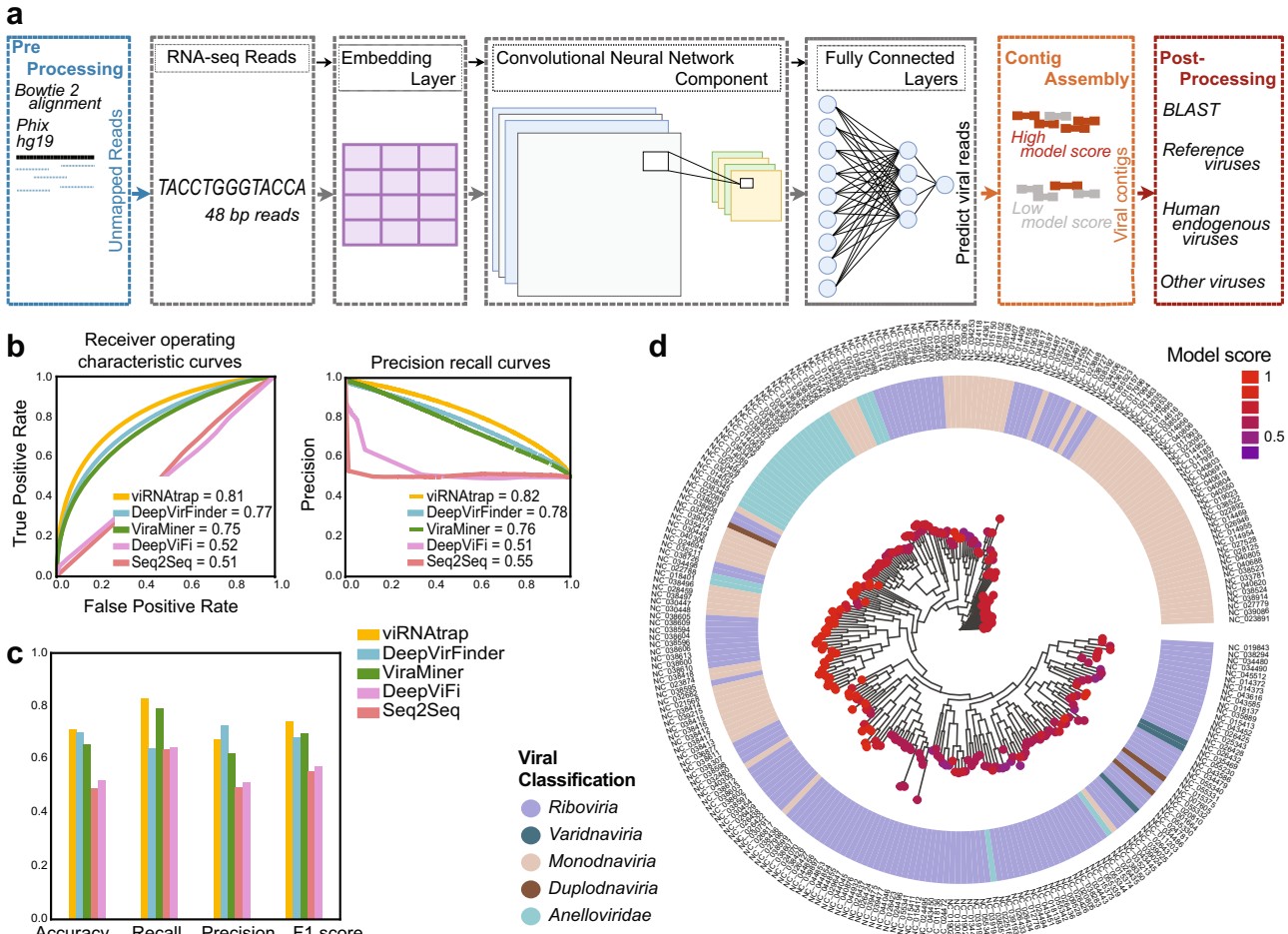

**Fig. 1 | Training and evaluation of the viRNAtrap framework. a** A schematic overview of the viRNAtrap framework. Unmapped reads were extracted and given as input to the neural network, to extract the viral reads and assemble viral contigs, that were compared against three viral databases using blastn. **b** Receiver operating characteristic and precision-recall curves showing the model performance when viRNAtrap and models used for comparison were applied to the test set. **c** Bar plots showing different metrics to evaluate the model performance for the test set, for viRNAtrap, and models used for comparison. **d** A phylogenetic tree showing the model scores for sequences from different human viruses with the respective virus classification (using the average assigned a score for each virus). Source data are provided as a Source Data file 1.

viRNAtrap pipeline. Notably, previous studies that systematically characterized viral presence across TCGA did not identify AAV2 in more than six LIHC samples[11,27], demonstrating the sensitivity of viR-NAtrap compared to other computational methods. We additionally detected AAV2 in one KIRC sample, one PAAD sample, and one matched non-cancer sample from LUAD (Fig. 2a).

We found several samples that express human polyomaviruses, especially polyomaviruses 6 and 7. Most notably, we found seven BRCA samples and two HNSC samples that express polyomaviruses. We additionally found Parvovirus B19 sequences in a few samples[29] (three cancer and one matched non-cancer); this virus has been mostly associated with normal tissues[30], but was also previously identified in isolated tumor cases[31,32]. We investigated possible genomic correlates of the expression of these viruses, including the tumor mutation burden (TMB, the rate of somatic mutations in a tumor, which is a biomarker and is annotated for all TCGA samples), and the chromosome-level aneuploidy (Methods). We found that HR-αHPV-positive samples have lower TMB and aneuploidy levels compared to HR-αHPV-negative samples (Fig. 2b). In contrast, LIHC cancer patients positive for HBV showed significantly higher TMB compared to HBV-negative samples (Supplementary Fig. 2). We additionally examined the association between the expression of known oncoviruses and overall survival. While none of the associations were significant after adjustment for

multiple hypotheses (Supplementary Fig. 2 and Supplementary Table 1), we found a trend that HR-αHPV-positive HNSC patients have better survival compared to HR-αHPV-negative patients (by the Kaplan–Meier curves Fig. 2c), which is confirmatory of previous studies[33,34]. We also found a positive association between the viral presence and the overall survival of LIHC patients with HBV (Supplementary Fig. 2 and Supplementary Table 1).

**Uncovering expression patterns of HERVs in cancer tissues**
To further demonstrate the utility of viRNAtrap, we analyzed the expression of HERVs across different tumor types in TCGA (HERVs were not used to train the viRNAtrap model). HERVs constitute ~8% of the human genome; most HERV sequences are remnants of ancestral retroviral infection that became fixed in the germline DNA[35,36]. HERV proteins are found expressed in different conditions including cancer tissues[37–41]. Specifically, the HERV-K family, which was most recently integrated into the human genome and is one of the most abundant HERV families in the human genome (along with HERV-H), was previously reported in tumor tissues and cell lines[42,43]. Moreover, recent findings reported the association between HERV expression and poor survival rates[12,36,44–46].

To comprehensively characterize HERV members that are expressed in different tumors, we established a database of potentially

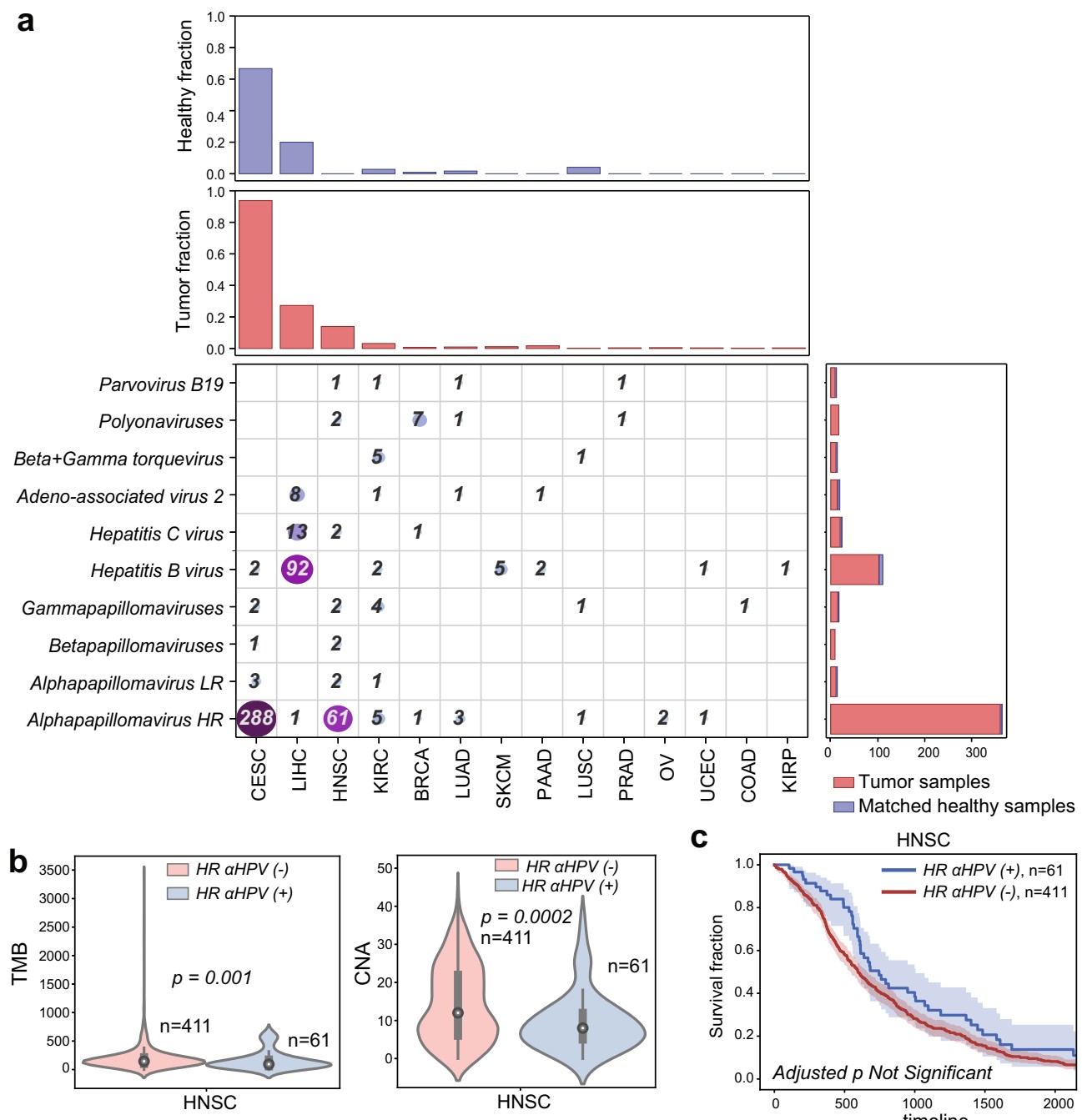

**Fig. 2 | Reference human viruses expressed in different tumor types. a** Heatmap showing the total number of virus-positive samples identified from RNA sequencing in different tumor tissues. The top panels show the fraction of tumor and non-cancer samples in which viruses were identified. The right panels show the number of viruses found in tumor and non-cancer samples. **b** Violin plots comparing the tumor mutation burden (TMB) and the number of chromosome-level copy number alteration (CNA) between HNSC patients where expression of high-risk alpha papillomaviruses was detected vs those patients where expression of high-risk alpha papillomaviruses was not detected. Black dots represent the medians, and the boundaries of the violin plots refer to the maximum and minimum values, respectively. Two-sided Wilcoxon rank-sum *p* value is reported. **c** Kaplan–Meier curves comparing the survival rates between HNSC patients where the expression of high-risk alpha papillomaviruses was detected (blue curve) vs those where the expression of high-risk alpha papillomaviruses was not detected (red curve). The FDR-adjusted two-sided log-rank *p* value is not significant (Supplementary Table 1). For Kaplan–Meier curves, shaded areas represent the confidence interval of survival. Source data are provided as a Source Data file 2.

functional HERVs that were extracted from the human genome (Methods). The viRNAtrap contigs were aligned against this database, to identify patterns of HERV expression in the 14 cancer types considered throughout this study.

As expected, we found that the most abundantly expressed HERV families are HERV-K and HERV-H. The fraction of samples expressing different individual HERV members was used to cluster tumor types. Interestingly, we found that squamous cell carcinomas (including cervical, lung, and head and neck) are clustered together based on the proportional distribution of expressed HERV members (Fig. 3a). The HERVs that are most abundantly expressed across different cancers include some that are in proximity to cancer-associated genes or single

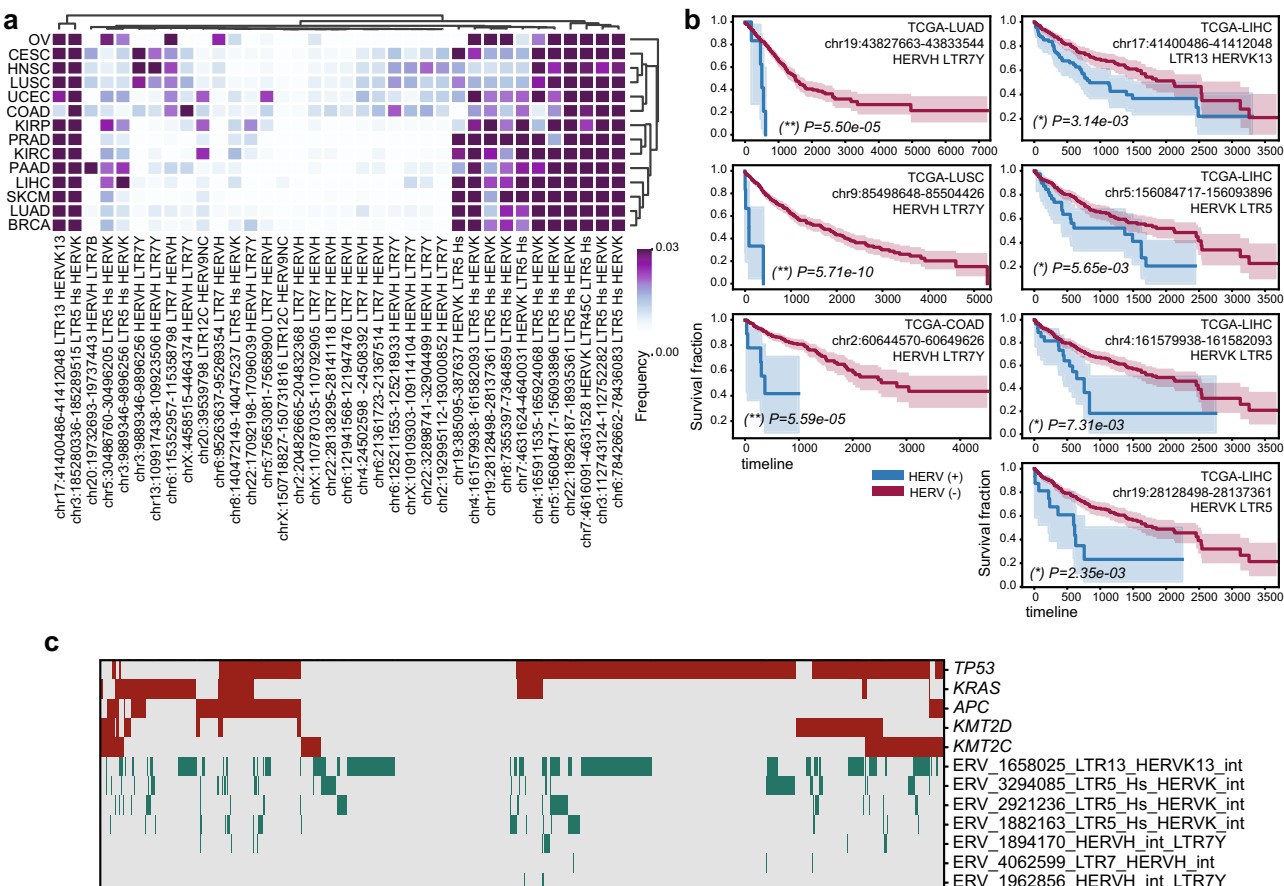

**Fig. 3 | Human endogenous retroviruses (HERVs) expressed in different cancer types. a** Heatmap clustogram clustering the proportion of HERVs across different tumor types. The rows are 14 TCGA tumor types. The 36 columns are the 36 distinct HERVs with the highest expression in human cancers, mapped to unique regions in the genome (Supplementary Data 5). **b** Selection of Kaplan–Meier curves comparing the survival rates between patients in which any HERV reads were detected (blue curves) versus those in which no HERV reads were detected (red curves). The unadjusted two-sided log-rank $p$ values are reported. (\*\*) global FDR $q$ < 0.05, (\*)

cancer-type specific FDR $q$ < 0.05. For Kaplan–Meier curves, shaded areas represent the confidence interval of survival. Additional significant associations between HERV and survival are reported in Supplementary Data 12. **c** Heatmap showing somatic mutations in major cancer driver genes (selected are the most frequently mutated driver genes in these samples, red) and the expression of HERVs that are significantly associated with survival in LIHC, LUAD, LUSC, and COAD (green). Source data are provided as a Source Data file 3.

nucleotide polymorphisms (SNPs) (Supplementary Data 3, 4). Specifically, one HERV-H member (chr2:204826665-204832368) is located 365 bp from the *ICOS* (Inducible T-cell costimulatory) gene, which has been associated with tumor immune responses[47–50]. In addition, one HERV9 member (chrX:150718827-150731816) is located 330 bp from the *PASD1* cancer/testis antigen gene (each of these two HERVs are found in ten TCGA samples, Supplementary Data 4, 5).

We investigated associations between HERV transcript presence and patients' overall survival (Fig. 3b). In agreement with previous studies[12,36,44–46], we find that patients with HERV-K- and HERV-H-positive cancer samples have significantly lower overall survival compared to HERV-K- and HERV-H-negative patients in COAD, LUSC, LUAD, and LIHC. Notably, every significant association that we identified between HERV presence and overall survival in these cancer types is negative (Fig. 3b and Supplementary Table 2).

To investigate the link between HERV expression and poor survival, we compared the TMB and aneuploidy scores between patients expressing HERVs and those without HERV expression. HERVs that were associated with poor survival were not associated with TMB or aneuploidy (Supplementary Data 6). We found that HERVs associated with poor overall survival were generally more likely to be expressed in the presence of somatic mutations in frequently mutated cancer driver genes, such as *TP53*, *KRAS*, *ARID1A*, and *PTEN* (using hypergeometric

enrichment, Supplementary Data 7). However, we did not find a strong association with mutations in any specific gene, and HERV expression was found even in samples with no somatic mutations in any of these genes (Fig. 3c and Supplementary Data 8)

### Finding divergent viruses in human cancer

We next investigated tumor expression of divergent viruses that have rarely or never been previously reported in human cancers. We aligned the contigs produced by viRNAtrap against a database of viruses (Methods) from different hosts that were not expected to be found in tumor tissues, including human, bat, mouse, insect, plant, and bacterial viruses. (Fig. 4a). We found multiple contigs of mosaic plant viruses in distinct samples from most tumor types, especially adenocarcinomas. For example, the watermelon mosaic virus was found in three colorectal cancer samples, and the Bermuda grass latent virus, which was previously reported in a COAD sample[10], was identified in multiple samples from three cancer types (COAD, LIHC, and UCEC; Fig. 4a). Mosaic plant viruses have been previously detected in human feces[51,52], which could suggest viral entry and travel through the digestive tract. However, it is unclear how mosaic plant viruses would reach other tumor tissues, such as the liver and the endometrium, and whether these are associated with an unidentified source of laboratory contamination.

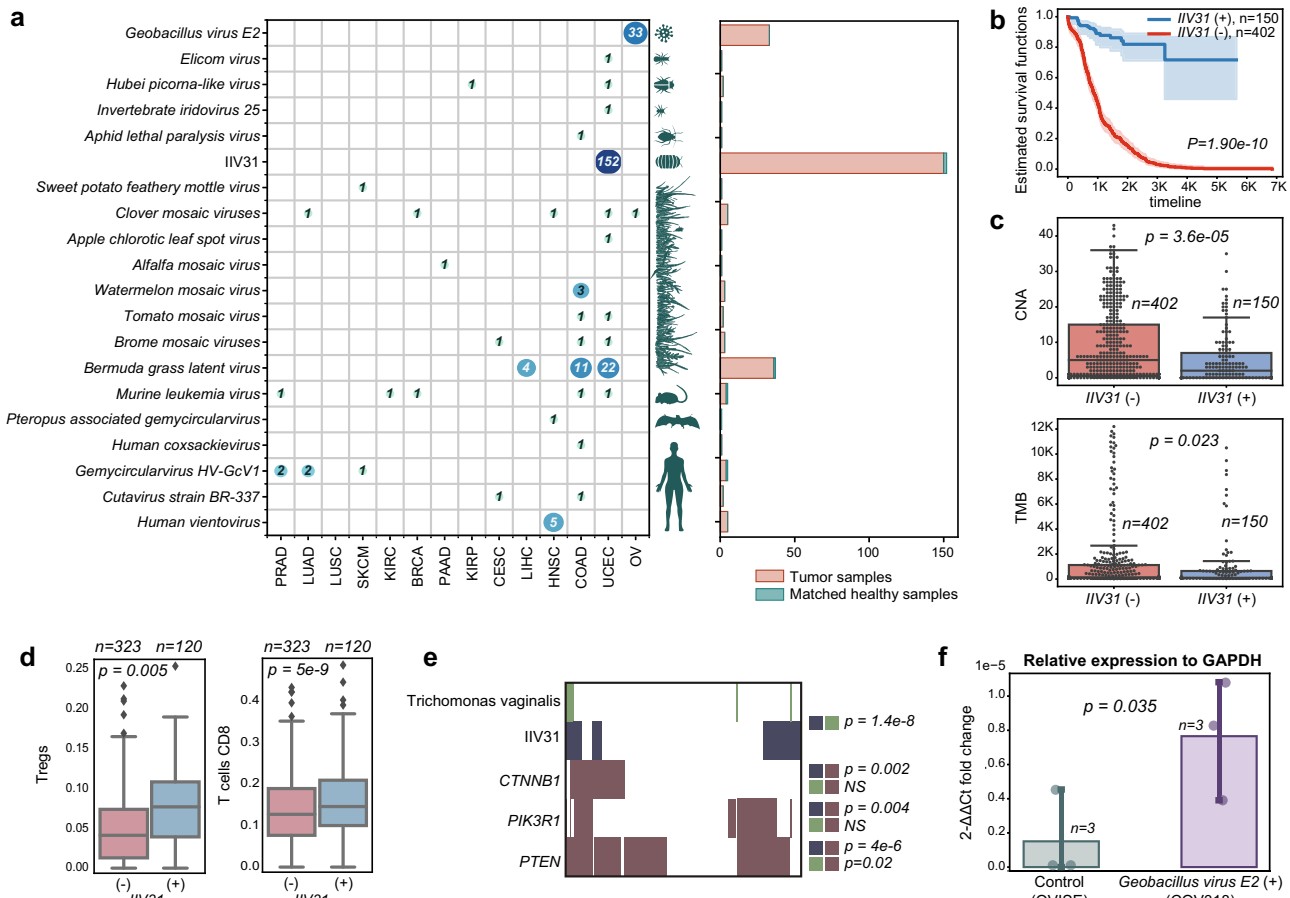

**Fig. 4 | Unexpected and divergent viruses infecting different host taxa across TCGA samples. a** Unexpected and divergent viruses expressed in TCGA samples. Each row in the matrix represents one virus and the entry in each column indicates the number of cancer samples of each type in which each virus was detected. The canonical hosts of each virus are depicted at the left of the matrix. At right, the aggregate number of tumor and normal samples containing reads of each virus are shown in a bar plot. **b** Kaplan–Meier curves comparing the survival rates between patients in which IIV31 reads were detected (blue curves) vs those where viral reads were not detected (red curves). For Kaplan–Meier curves, shaded areas represent the confidence interval of survival. The log-rank $p$ value is reported. **c** Box plots comparing the chromosome-level copy number alteration (CNA, top panel) and the tumor mutation burden (TMB, bottom panel) between cancer patients where IIV31 is found (blue) and patients where IIV31 is not found (red). Two-sided Wilcoxon rank-sum $p$ value is reported. **d** Box plots comparing CIBERSORT-inferred proportions of regulatory T cells (Tregs) and CD8 T cells between patients positive and negative for IIV31. Boxes show the quartiles (0.25 and 0.75) of the data, center lines show the medians, and whiskers show the rest of the distribution except for outliers. A two-sided Wilcoxon rank-sum $p$ value is reported for comparisons assigned with FDR $q < 0.05$. **e** *Trichomonas vaginalis* and somatic mutations in *PTEN, CTNNB1*, and *PIK3R1* are associated with IIV31 presence. One-sided Fisher's exact test $p$ values are provided. **f** Bar plot comparing the fold change (relative to GAPDH) between the COV318 cell line that was predicted as *Geobacillus*-positive, and the OVISE cell line that was used as control. Error bars show the standard deviation. The one-sided $t$-test $p$ value is provided. Source data are provided as a Source Data file 4.

Notably, we identified expression in five head and neck carcinoma samples of a *Vientovirus*, a member of the recently characterized human virus family *Redondoviridae* that is associated with the human oro-respiratory tract[53] (Fig. 4a and Supplementary Data 3, 9). We also found expression of a *Gemycircularvirus* HV-GcV1[54] in distinct samples from several cancer types, and *Cutavirus* expression in one COAD and one CESC sample each. We additionally detected human coxsackievirus[55] in a COAD sample, confirming a previous report[10].

We also found expression of a few arthropod viruses in TCGA, almost exclusively in UCEC samples (Fig. 4a), the most notable of which is *Armadillidium vulgare* iridescent virus (IIV31)[56]. We detected reads that align with IIV31 proteins in 152 endometrial cancer samples (which constitute more than 25% of endometrial cancer samples studied). While we did not find previous reports of IIV31 in these samples, reads that align to the same strain were recently detected in a few DNA sequencing samples, but were filtered because these were not included in databases of multiple pipelines[12]. IIV31 is in *Betairidovirinae*; members of this subfamily of dsDNA viruses infect a wide variety of arthropods, including common insect parasites of humans[57]. One

study speculated on the role of *Betairidovirinae* transmitted by mosquitos in human disease[58], but, to our knowledge, their presence in humans has not been reported before. While *Betairidovirinae* are not considered to be pathogens of vertebrates, one study showed that the model *Betairidovirinae* insect iridovirus 6 (IIV6) was lethal to mice after injection, while heat-inactivated IIV6 was not[59]. Additional studies have shown that *Betairidovirinae* can infect vertebrate predators of infected insects as well as several vertebrate cell lines[60]. Therefore, *Betairidovirinae* may opportunistically infect vertebrates, including humans.

We identified different IIV31 genes expressed in UCEC samples, and samples positive for IIV31 proteins originate from different batches and sequencing centers (Supplementary Data 10). In addition, we found that IIV31 presence was strongly and positively associated with overall survival (Fig. 4b), and negatively associated with TMB and chromosome-level aneuploidy (Fig. 4c, d). We did not identify a path to contamination by IIV31; the multiple origins of IIV31-positive samples and significant associations between IIV31 expression and other cancer properties both suggest that IIV31 is not a contaminant. Of the most highly expressed IIV31 proteins, we found an IAP apoptosis inhibitor

homolog and serine/threonine protein kinases that were individually associated with poor overall survival (YP_009046765, YP_009046752, and YP_009046774, respectively), as well as a *RAD50* homolog (YP_009046808, Supplementary Fig. 3 and Supplementary Data 10).

We found a significant positive association between IIV31 and CIBERSORT[61] inferred CD8+ T-cell frequency and Treg frequency (Fig. 4d). These findings, together with the association with improved survival, suggest that IIV31 could be linked with a different infection, either directly or indirectly. We explored the association of IIV31 infection with *Trichomonas vaginalis* (TV)[62] infection. TV is a single-celled protozoan pathogen that infects the human urogenital tract[63], and has been associated with an increased risk of cervical cancer, which is enhanced by HPV coinfection[64]. We found that TV is expressed in multiple UCEC tumor samples (we verified 21 TV-positive tumors with strict alignment parameters, due to a high false positive rate when aligning against TV transcripts). Indeed, TV-positive samples are highly enriched with IIV31-positive samples (Fisher exact test $p$ value = 1.4e-8). Both TV and IIV31 are significantly associated with somatic *PTEN* mutations, which are linked to better survival in endometrial cancers[65] (whereas the presence of IIV31 is also associated with mutations in *CTNNB1* and *PIK3R1*, Fig. 4e).

We additionally identified *Geobacillus* virus E2 expression in 33 ovarian cancer samples; this virus is likely the most frequently expressed virus in high-grade serous ovarian cancer. To further validate the presence of the *Geobacillus* virus E2, we applied viRNAtrap to cell line data from CCLE[66]. We identified the COV318 cell line as *Geobacillus* virus E2-positive and identified the OVISE cell line as a virus-negative control. Through qRT-PCR we validated the expression E2 in the predicted-positive cell line COV318 (Fig. 4f). These results verify that *Geobacillus* virus E2, which was never found in ovarian cancer before, is indeed expressed in ovarian cancer cells, and that viRNAtrap can be used to sensitively detect virus-positive samples. *Geobacillus* bacteria has been previously detected in multiple ovarian cancer samples[67,68]. While we could not pinpoint the *Geobacillus* species harboring the phage, it is likely within those previously found in ovarian cancer samples[67,68].

We found murine leukemia virus[69] expression in distinct samples from five cancer types. However, murine leukemia virus contamination has been reported for cell culture due to human DNA preparation[70]. Our method additionally detected a previously unknown virus in a matched non-cancer sample from one HNSC patient, with protein similarity to *Pteropus* (fruit bat)-associated *Gemycircularvirus* and several other gemycircularviruses (Supplementary Data 3, 9).

## Discussion

Identification of viruses from tumor RNA sequencing allows for the potential discovery of new carcinogenic agents and mechanisms. The discovery of novel and divergent viral species that contribute to cancer initiation and progression is crucial for the development of new therapeutics, including vaccinations, screening practices, and antimicrobial treatments. Viruses are currently identified from sequencing reads based on similarity to known viruses[71]. However, when studying viruses from short reads, typical with Illumina-based RNA sequencing, reads originating from divergent viruses may share little sequence similarity to known viruses, rendering the identification of novel viruses highly challenging.

To address this challenge, we developed viRNAtrap, a new, alignment-free framework to identify viral reads from RNAseq and assemble viral contigs. The contigs detected by viRNAtrap can be aligned to different viral databases, as we demonstrate in this study, to rapidly identify viral expressions of interest in tumor samples. We curate a database of HERVs that comprise intact retroviral genes in the human genome and survey the expression of these viruses across different cancer tissues. Through a database of divergent viruses, we demonstrate that viRNAtrap identifies viruses in TCGA samples that

were not detected in previous studies. This is enabled through an integrative method that uses the model scores to assemble viral reads rather than aligning short divergent reads to viral databases or applying assembly to many unmapped reads. We further show that using the deep learning model substantially improves the running time, while not compromising sensitivity if more than five viral reads are present (Supplementary Fig. 4, see Methods). Importantly, the output of viRNAtrap can be alternatively used as input to motif search tools, to potentially identify highly divergent viruses. Because the deep learning model underlying viRNAtrap was trained to distinguish viral from human sequences, the model predictions for sequences derived from a range of other organisms is not defined. Future work could train models to identify viruses from a variety of other organisms, and, with the viRNAtrap framework, achieve higher sensitivity for viral detection.

We employ viRNAtrap for exploratory data analysis and characterize viruses that are expressed across 14 cancer tissues from TCGA and analyze their genomic and survival correlates. Interestingly, while the expression of some exogenous cancer viruses is known to be associated with improved survival, we found that the expression of human endogenous viruses is strictly associated with poor survival rates. Expression of a virus of the subfamily *Betairidovirinae*, which are pathogens of insects, found in endometrial cancer tissues was similarly associated with significantly better overall patient survival. For all divergent viruses reported in this study, the presence and classification of multiple viral reads was verified by targeted blastn- and blastx-based sequence analyses in different samples. However, it is not possible to model all contaminants of viruses that may have infected the samples during laboratory procedures[16].

Perhaps, the most interesting divergent virus we found is IIV31 from the subfamily *Betairidovirinae*, which was frequently detected in UCEC TCGA samples. Interestingly, IIV6, a very close relative of IIV31, can infect a variety of vertebrates including mice, and induces an immune response in mammalian tissues[60,72]. Thus, one possibility is that IIV31 is transmitted to the uterus through another insect, such as the crab louse. While we have not yet confirmed the source of this virus, our results imply that its presence may be a direct or indirect consequence of *Trichomonas vaginalis* infection. Therefore, it shows that viRNAtrap is sufficiently powerful to identify a previously unknown viral transcript in tumor samples, whether oncogenic or neutral. Through this analysis, we also identified TV reads in multiple endometrial cancer samples, indicating a possible new association between TV and endometrial cancer, like the known association of TV with cervical cancer[64]. One of the established pathogenic mechanisms of TV infection in humans, which may also explain the frequent HPV coinfection, is that TV secretes exosomes that have the effect of suppressing CXCL8[73]. Interestingly, low expression of CXCL8, like infection with TV, has been associated with a favorable prognosis in cervical cancer[74]. Thus, it is possible that the presence of IIV31 is a secondary infection in patients already infected with TV or some other pathogen that suppresses the human anti-viral response.

Importantly, we identified the *E2 Geobacillus* virus in 10% of high-grade, serous ovarian cancers, making it the most frequently expressed virus in this cancer type. We experimentally verified that *E2 Geobacillus* is indeed expressed in cell lines. We also found expression of a *Redondoviridae* member in head and neck cancers that was not previously reported[75]. This finding calls for a study of the role of *Redondoviridae* in tumor initiation and progression, as this family of viruses was only recently detected in humans and associated with different clinical conditions.

In conclusion, we developed viRNAtrap, a new software for alignment-free identification of viruses from RNAseq, allowing rapid characterization of viral expression and detection of divergent viruses. We applied it to tumor tissues from TCGA, uncovering expression patterns of different groups of viruses. We report previously unrecognized associations between several forms of cancer and several

unexpected viral clades, including viral clades canonically found in produce and in insect parasites of humans. Future studies may employ viRNAtrap to find viruses that contribute to other malignancies.

## Methods

### Training a neural network to distinguish viral RNA sequencing reads

The viRNAtrap framework is composed of two main components, illustrated in Fig. 1a. The first is a deep learning model, which was trained to accurately distinguish viral from human reads using RNA sequencing. The second assembles the predicted viral reads into contigs. The trained neural network is composed of one 1D-convolutional layer and three fully connected layers, one of which is the final output layer. The RNA sequences were one-hot encoded to vectors that were given as input to the model. The learning rate was set to 0.0005, we used 64 filters with ReLU as an activation function in the convolutional layer, followed by one pooling layer for feature extraction. The global extracted features from the convolutional layer are passed to three fully connected layers, to make a prediction based on a sigmoid activation function in the output layer.

To train the model, we collected human and viral sequencing data. Coding sequences of human and other placental viruses were downloaded from the Virus Variation Resource[76]. Human transcripts for hg19 were downloaded from NCBI Human Genome Resources[77]. These sequences were segmented into 48 bp segments, which is the read length for the RNAseq in almost all tumor types in TCGA; only a few tumor types that were added chronologically last to TCGA used longer reads. We used a 48 bp window size for human transcripts and a 2 bp window size for viral sequences, to balance the positive and negative data. Then, these were randomly split (where all segments of each transcript were considered together) into balanced train, validation, and test sets ($n = 8,000,000$, 800,000, and 2,558,044, respectively).

### Model performance evaluation and comparison to existing methods

We evaluated the performance of the model using the area under the receiver operating characteristic curve (AUROC), the area under the precision-recall curve (AUPRC), as well as accuracy, precision, recall, and F1-score, for the test dataset. We trained multiple models with different architectures and hyperparameters and then selected the model with the highest average between the validation-set AUROC and recall. The model was trained using TensorFlow 2.6.0 and Keras[78]. We compared the performance of our model to models from DeepViFi[16], DeepVirFinder[15], ViraMiner[21], and off-the-shelf Seq2Seq model. Because this is the first approach trained to predict viruses from RNA sequencing reads of length 48 bp, we used our training data to retrain each of these models, following the instructions provided by each method, and evaluated the AUROC, AUPRC, accuracy, precision, recall, and F1-score using our test set (see Supplementary Methods for a detailed description of hyperparameters used). Importantly, existing methods were not designed for reads shorter than 150 bp, therefore they should not be expected to perform as well as viRNAtrap on 48 bp segments, for which viRNAtrap was optimized. Our comparison does not rule out the possibility that new hyperparameter optimization for this purpose may enhance the performance of existing methods for 48 bp sequences.

### Assembling viral contigs from neural network predicted viral reads

Once the viRNAtrap model predicts the probability of a viral origin of each read, reads with model scores more than 0.7 are used as seeds to assemble viral contigs. Viral contigs are assembled using an iterative search for substrings with exact matches between 24 bp k-mers. Each seed is complemented from the left and right ends using its left-most and right-most 24 bp k-mers. For both the left and right assembly,

reads containing the left or right-most k-mers in a different position from the read that is being searched are identified. The read adding the maximal number of bases to the assembled contig is used to complement the left and right contigs. The model scores that were assigned to reads that are used to assemble each contig were averaged, and the assembly terminates if the average score is below 0.5. Finally, the right and left contigs are concatenated, to yield a complete viral contig. This algorithm was implemented in Python 3 and subsequently in C, which improved the running time by more than an order of magnitude for inputs with large numbers of reads.

### Data pre-processing

We downloaded RNA sequencing data from Genomic Data Commons (GDC; https://portal.gdc.cancer.gov/)[79] as BAM files. High-quality reads were selected and mapped with Bowtie2 against hg19 (1000 Genomes version) and PhiX phage (NC_001422), and only the unmapped reads were kept. Then, we merged the paired-end reads and converted them to fastq files, which were used as input for the viRNAtrap framework, to yield predicted viral contigs.

### Viral databases

Viral contigs yielded by the assembly component were used as inputs to blastn[24]. Three databases were used to search for viruses (with an E-value threshold of 0.01):

(1) RefSeq reference human viruses, downloaded from the National Center for Biotechnology Information (NCBI)[77], to which we added human papillomaviruses strains that are not in RefSeq from PAVE (https://pave.niaid.nih.gov)[80]. Reference viruses were searched using blastn, with default parameters except for a word size of 15 (lower than the default of 28), which was chosen to allow identification from short contigs.

(2) more divergent viruses were obtained from RVDB[81] (https://hive.biochemistry.gwu.edu/rvdb/) which was then filtered to remove non-viral elements, endogenous viruses, and accessions that were consistently not verified using blastn against the nonredundant (nr) blast nucleotide database.

(3) Human endogenous viruses. We curated a database of potentially functional HERVs through the evaluation of viral protein completeness (in contrast to a previous study that evaluated HERV expression in distinct RNAseq datasets[82]). The initial genomic locations of reported HERV elements were downloaded from the HERVd HERV annotation database (https://herv.img.cas.cz)[83]. The nucleotide sequences in hg19 for each reported HERV were extracted using twoBitToFa[84]. We then applied blastx against NR with an E-value cutoff of 1E-4, as well as a profile search[85] against collected POL proteins, where the profile was obtained by collecting POL genes annotated in GenBank in lentiviruses (as of September 2016) and aligning their amino acid sequences using MAFFT[86]. Sequences with at least one identified retroviral protein motif of POL/RT, GAG, or ENV were extracted, yielding 3044 HERVs that were considered for search in TCGA samples (Supplementary Data 5). Importantly, the high mutation rate of HERV[87] prohibits most HERV sequences from aligning to the human genome in pre-processing[12,88], however, in rare cases, HERV regions that are conserved would not be identified by this approach.

### Quality standards for virus identification

For all viruses, blastn was applied with an E-value cutoff of 0.01 and any sequences with a match to contaminant accessions (that were associated with vector contamination) were filtered out.

a. Reference viruses. For every sample, contigs mapped to each accession were extracted. Identified accessions with maximum qcov across contigs of more than 90%, average qcov of more than 50%, and average similarity of more than 90% were considered. Accessions with maximal contig length under 100 bp were manually inspected and verified against nr.

b. Human endogenous viruses. For every sample, contigs mapped to each HERV were extracted. HERVs with contigs longer than 200 bp, and with average qcov and similarity of more than 95% were considered.

c. Divergent viruses. For every sample, contigs mapped to each accession were extracted. Viruses already identified through the reference database were removed. Identified accessions with maximal contig length of more than 300 bp and qcov of more than 40%, or with maximal contig length of more than 100 bp and qcov of more than 75% and average similarity of more than 75% were considered for manual inspection.

All instances of divergent viruses identified in TCGA samples were verified using blastn against nr, to support that the virus strain is indeed the best match to a viral contig generated by viRNAtrap. We reason that non-reference viruses (divergent viruses and viruses of non-human hosts) that were identified and verified in more than one sample were less likely to be a contaminant or isolated events, whereas samples with fewer reads from such viruses may be filtered due to the strict filtering. We therefore additionally searched using the STAR aligner[89] across tumor types where these viruses were identified through viRNAtrap (Supplementary Data 3). The following accessions were additionally searched using STAR to increase sample coverage (as these were the most interesting divergent strains found across multiple samples): Bermuda grass latent virus (NC_032405), *Armadillidium vulgare* iridescent virus IIV31 (NC_024451), *Geobacillus* virus (NC_009552), and the human lung-associated vientovirus (NC_055523)

### Filtering contaminants
To filter vector contaminants, we applied VecScreen[90] to the assembled contigs that have been mapped to viruses through our databases, where virus accessions associated with vector contaminants were entirely removed from the search (Supplementary Data 11).

In addition, we examined the application of software such as Kraken2[91] to the RNAseq reads for filtering reads that are not likely of viral origin, by applying Kraken2 to reads of LIHC samples. However, we found that 99% of the reads would not be filtered using this approach (Supplementary Fig. 5), likely due to the short reads (48 bp) for which Kraken has not been designed or evaluated, as longer sequences are known to be more accurately mapped[92].

### Genomic correlates of viral expression
We correlated viral expression with genomic markers across TCGA samples. Chromosomal aneuploidy levels for TCGA samples were extracted from[93] and the total number of chromosome-arm-level alterations was used. The tumor mutation burden was defined to be the total number of somatic mutations in each sample, downloaded from the Xena browser[94] (https://xenabrowser.net). CIBERSORT[61] software was applied to TCGA samples using the default set of 22 immune-cell signatures.

### Cells and culture conditions
Human ovarian cancer cell lines COV318 and OVISE were cultured in RPMI1640 medium containing 10% fetal bovine serum (FBS) and 1% penicillin-streptomycin under 5% $CO_2$. All of the cell lines were authenticated at The Wistar Institute's Genomics Facility using short-tandem-repeat DNA profiling. Regular mycoplasma testing was performed using a LookOut mycoplasma PCR detection kit (Sigma, cat. no. MP0035).

### Experimental validation of the *Geobacillus* virus E2 in ovarian cancer cell lines
Reverse-transcriptase qPCR (RT-qPCR) RNA was extracted using TRIzol reagent (Invitrogen, cat. no. 15596026). Extracted RNA was used for reverse-transcriptase PCR using a High-capacity cDNA reverse transcription kit (Thermo Fisher, cat. no. 4368814). Quantitative PCR was performed using a QuantStudio 3 real-time PCR system. GAPDH was used as an internal control. The fold change was calculated using the 2-ΔΔCt method. The primers used for reverse-transcriptase qPCR are: GAPDH forward, GTCTCCTCTGACTTCAACAGCG and reverse, ACCACCCTGTTGCTGTAGTAGCCAA; *Geobacillus* virus E2 terminase forward, TTGCGATGCGTACTCAGACT and reverse, CTCTTTTTGGTCAGCAGCGG Primers were obtained using NCBI primer design tool as shown in the Supplementary Information. The primers were synthesized by Integrated DNA Technologies IDT. A specification of the primer design is provided in the Supplementary Information.

### Identification of *Trichomonas vaginalis*-positive samples
UCEC unmapped (to hg19) reads were aligned to the reference genome of *Trichomonas vaginalis* (GCF_000002825)[62] strain G3 using blastn[24] with E-value <1e-8 and more than 90% identity. These thresholds were set to remove false positives that were frequent when aligning against *Trichomonas vaginalis* when examining both blastn[24] and STAR aligner[89]. TV reads for each TV-positive sample were verified by manual inspection of the output alignments.

### viRNAtrap performance evaluation
To evaluate the contribution of the model to the viRNAtrap pipeline we re-ran viRNAtrap on 10 LIHC samples, and additionally ran a modified viRNAtrap pipeline not using the model, on the same system. We compared the viruses identified by the model-based approach to those that are identified when the pipeline is applied without using the model (Supplementary Table 2, showing similar viruses with a different number of contigs). We additionally compared the running time of the two approaches (Supplementary Fig. 4).

To evaluate the sensitivity of the viRNAtrap pipeline based on the number of viral reads present in a sample, we performed a simulated analysis. From the test dataset, we downsampled groups of viral reads with different group sizes (10,000 groups for each size, from one read up to 10 reads), and we evaluated the number of groups with at least one read that is scored above 0.7, which is the seed threshold used for the viRNAtrap assembly. Therefore, this analysis is estimating the probability of identifying viruses based on the number of reads present. We found 93 and 99% of the groups with more than 5 and 9 reads, respectively, would be identified.

### Statistical methods
Survival analyse, including Kaplan–Meier curves plots and log-rank test $p$ values, were obtained using the Python lifelines package (v0.26.4)[95]. $P$ values comparing TMB and aneuploidy between two groups were computed with two-sided Wilcoxon rank-sum tests. Heatmap clustograms were generated through seaborn clustermap.

Viruses with significant log-rank $p$ values are reported as significantly associated with survival.

None of the reference viruses were significantly associated with survival after FDR correction (Supplementary Table 1), however, we report in Fig. 2 the association between HR-HPV with unadjusted $p$ value because it is confirmatory of a known association between HR-HPV and HNSC survival[33,34].

For HERV, our exploratory data analysis uncovered some significant associations with complete hypothesis testing. We present in the main text selected associations with at least five cases in each group. Nevertheless, FDR correction was applied within each cancer type for all HERV associations, and we additionally applied a global FDR correction for all comparisons across cancer types, yielding some significant associations with less than five positive cases. The complete significant associations between survival and viral presence are reported in Supplementary Data 12.

**Reporting summary**

Further information on research design is available in the Nature Portfolio Reporting Summary linked to this article.

## Data availability

The complete training and test data as well as viral databases generated in this study have been deposited in the Zenodo database under the accession code https://doi.org/10.5281/zenodo.7548375. The results shown here are in whole or part based upon data generated by the TCGA Research Network: https://www.cancer.gov/tcga. The raw FASTQ RNA sequencing data are protected and are not publicly available due to data privacy laws, but are available under restricted access as data can be unique to an individual. Access can be obtained from the Genome Data Commons (GDC) after receiving permission via dbGaP, following the steps described in: https://www.ncbi.nlm.nih.gov/projects/gap/cgi-bin/study.cgi?study_id = phs000178.v11.p8. The processed data including viruses identified and respective statistics are available as supplementary Data 3. The complete data generated in this study are provided in the Supplementary Information/Source Data file. Source data are provided with this paper.

## Code availability

The scripts for pre and post-processing and the viRNAtrap package are available through GitHub: https://github.com/AuslanderLab/virnatrap and Zenodo under accession code: https://doi.org/10.5281/zenodo.7548375.

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

## Acknowledgements

The research reported in this publication was supported in part by the National Cancer Institute of the National Institutes of Health under Award Number R00CA252025 (N.A.), RF1-AG063481, P30-CA016520 (Q.L.), and NIH RO1 AI153508, Commonwealth of Pennsylvania SAP# 4100089371, and P30 CA010815 (P.M.L), and by the Intramural Research Program of the National Institutes of Health, National Cancer Institute (A.A.S.).

## Author contributions

N.A. initiated the project. Q.L., R.Z., A.A.S., and N.A. supervised work. A.E., X.H., R.Z., A.A.S., and N.A. designed and performed experiments and analyses. A.E., Y.Y., D.E.S., J.W., A.A.S., and N.A. wrote and tested software. P.M.L and Q.M. contributed to data interpretation and exploratory analyses. K.T. and Q.L. revised the survival analysis.

## Competing interests

P.M.L. is a founder of and advisor to Vironika, LLC. All other authors declare no competing interests.
