## [Peer Review File · Nature Communications]

REVIEWER COMMENTS

Reviewer #1 (Remarks to the Author):

Elbasir and colleagues introduce viRNAtrap a pipeline that uses a deep learning model to identify viral reads and assembles them into contigs. The pipeline is then applied on the non-human reads from tcga samples. The authors then do descriptive analysis on virus occurrence and well as clinical progression and survival analyses on selected viral families. While the manuscript covers a highly important question and provides a nice connection of method development and large data analysis, it leaves several questions unanswered.

Major:

A1) Methodological Novelty

Vineet Bafna's lab recently introduced a closely related tool DeepViFi (<https://dl.acm.org/doi/pdf/10.1145/3535508.3545551>) and compared it to ViraMiner, DeepVirFinder, ViFi, and an off-the-shelf seq2seq mode.

Such a comparison, e.g. on the data of the benchmark provided in that paper would be very helpful to judge whether there is indeed a substantial in performance.

Further, it should be acknowledged that not only bacteria infecting viral reads have been classified using deep learning, but also human infecting viral reads have been classified using machine learning models earlier, e.g HIVF (<https://doi.org/10.1111/tbed.13314>) or DeePac (<https://doi.org/10.1093/nargab/lqab004>).

A2) Influence of contaminants / false positive viral identifications

It can by no means be expected that the non-human and PhiX reads are all of viral origin, but many will be bacterial reads (and their phages), fungi etc. same as possible index swapping artefacts or run carryovers of Illumina machines. There are multiple approaches available to account for that on methodological level in the learning approaches, e.g., deepViFi has an open set approach, DeePac uses multiclass learning to distinguish them, but even more so quick a kraken (or similar) analysis of these reads to filter out non-likely viral reads appears appropriate before actually running viRNAtrap to ensure that no reads of other origin contaminate the viral assembly and lead to potential false positive viral identifications.

Further, I could not find sufficient information on the quality standards for the viral assembly and blast analysis. What was the minimal contig length considered? How homogeneous was the coverage on the contigs? What were the standards of the blast analysis? Why were non-reference

viruses only deeper investigated when occurring in more than one sample? It would be very helpful if the authors provided quality information on the assembly for each identified virus.

A3) The analysis of the clinical outcome is interesting, but not entirely novel, e.g. <https://doi.org/10.3390/v12090956> studied HERV expression in tcga and their impact on survival. Surprisingly, this manuscript is cited with regard to the ancestry of HERV, but not in the context of survival (while it basically addresses the same question as the authors do, the authors claim that the impact of HERVs on cancer progression and clinical outcomes is not well understood). It would be helpful if the authors clarified these similarities

Minor:

B1) The software is not available as claimed by the authors in the Code and Software Submission Checklist. There was no demo on the authors' data available and scripts for reproduction which are claimed to be available in the submission checklist, are not given, rather the manuscript states they will be made available after publication (why not for review now?). Typical installation or run times are not given as stated.

B2) Similarly, it needs to be clarified how viral databases will be accessible (without restrictions) as they are currently not provided.

B3) I could not find the word document specifying the primer design in the submission

B4) Multiple Testing. The statistical methods section does not state that any multiple testing correction was performed. Although the manuscript (e.g. in figure 2) only shows a limited number of tests, these include some rare choices of viruses (e.g. torquevirus) making me assume that the authors tested all viruses on all cancer types (or later on all HERVs etc.). Most p-values shown in figure 2 for instance are not significant any longer after Bonferroni correction for the roughly 50 observed (and 140 theoretically possible) combinations of cancer tissues and viral families. There are smarter ways to do multiplicity correction than Bonferroni and maybe the authors have some (unstated) assumptions to reduce the hypothesis space, but with the current information provided these findings should not be considered statistically significant. Further, there should be a full reporting of all results (e.g. in the supplement) not only those chosen for discussion.

Reviewer #2 (Remarks to the Author):

The authors present a novel pipeline to analyse viral reads in human RNA sequencing data sets. Based on a two step approach they first use a deep learning model trained on human and viral reads to identify potential viral reads. In a second step these reads are assembled using a simple overlap based approach. The resulting contigs are then analysed with blastn to identify viral sequences.

Based on the data presented it is difficult to assess the performance of their approach as a description of the read support for the individual viral detections is missing.

Major

- The authors seem to only present virus positivity (1) or negativity (0) in the supplementary tables 3,4 and 9. For an evaluation of virus positivity the number of supporting reads would be very helpful.
- The authors could compare their approach to a simple assembly of the extracted reads and analysis of the resulting contigs. This would enable to evaluate whether they loose virus supporting reads and thereby sensitivity by their approach and enable to evaluate the speed improvement achieved.
- The authors restrict their analysis to the reads not mapped to the human reference genome. It is unclear to me how this affects the detection of HERVs that are present in the human reference genome and should therefore not be detected in an optimal way.

Minor

- Did the authors correct for multiple testing throughout the manuscript when testing for associations e.g. across entities or mutated genes

We thank the two reviewers for their insightful comments and suggestions on our manuscript (NCOMMS-22-33903). We have addressed all comments and believe the revisions have substantially improved this work and helped further support the utility of our tool viRNAtrap. Please see below a point-by-point response describing how we have revised and improved our manuscript. References cited in the response to either reviewer appear together after the responses to reviewer 2. If a reference is cited in text that also appears in the manuscript, the citation number in this document corresponds to the reference list in this document and the citation number is different in the manuscript. All text changes related to any reviewer comment are in red font, even minor changes. Some minor rewording and reformatting changes unrelated to the reviewer comments remain in black font.

Reviewer #1 (Remarks to the Author):

Elbasir and colleagues introduce viRNAtrap a pipeline that uses a deep learning model to identify viral reads and assembles them into contigs. The pipeline is then applied on the non-human reads from tcga samples. The authors than do descriptive analysis on virus occurrence and well as clinical progression and survival analyses on selected viral families. While the manuscript covers a highly important question and provides a nice connection of method development and large data analysis, it leaves several questions unanswered.

Major:

A1) Methodological Novelty

Vineet Bafna's lab recently introduced a closely related tool DeepViFi

(<https://dl.acm.org/doi/pdf/10.1145/3535508.3545551>) and compared it to ViraMiner,

DeepVirFinder, ViFi, and an off-the-shelf seq2seq mode.

Such a comparison, e.g. on the data of the benchmark provided in that paper would be very helpful to judge whether there is indeed a substantial in performance.

We thank reviewer for this comment. We completely agree that a comprehensive benchmarking comparison would further support the need for our viRNAtrap method. To this

end, we compared the performance of viRNAtrap to existing methods as the reviewer suggested. The methods to which we compared against are: DeepViFi, ViraMiner, DeepVirFinder and what was named 'off-the-shelf seq2seq' in the DeepViFi paper. The other methods were retrained using our training set and evaluated using our test set. Importantly, viRNAtrap outperforms all trained predictors in distinguishing viral from human reads, based on accuracy, ROC AUC, AUPR, recall and F1 score. DeepVirFinder outperformed all methods only on precision. viRNAtrap is trained to optimize the recall, because the true viruses captured are most important when searching for viruses in cancer, and when alignment-based methods are used to further validate any positives

We now report the results of this analysis in the revised Results section and describe how we did the comparison in the revised Methods subsection entitled "Model performance evaluation and comparison to existing methods"

"We evaluated the performance of the model using the Area Under the Receiver Operating Characteristic Curve (AUROC), the Area Under the Precision Recall Curve (AUPRC), as well as accuracy, precision, recall, and F1-score, for the test dataset. We trained multiple models with different architectures and hyperparameters and then selected the model with highest average between the validation-set AUROC and recall. The model was trained using TensorFlow 2.6.0 and Keras¹. We compared the performance of our model to models from DeepViFi², DeepVirFinder³, ViraMiner⁴ and off-the-shelf Seq2Seq model. Because this is the first approach trained to predict viruses from RNA sequencing reads of length 48bp, we used our training data to retrain each of these models, following the instructions provided by each method, and evaluated the AUROC, AUPRC, accuracy, precision, recall, and F1-score using our test set."

We believe that one reason that viRNAtrap model outperforms these leading approaches is that it is the only model that was designed and optimized for short RNAseq reads of length 48bp in model selection and hyper-parameter optimization. The other methods, which the reviewer suggested to compare against, were originally designed for other sequencing technologies that are not associated with such short reads. This is now mentioned where these findings are described in the revised results section:

“We compared the performance of this model to previous models trained to identify viruses, namely DeepViFi², DeepVirFinder³, ViraMiner⁴, as well as a method called ‘off-the-shelf Seq2Seq’ compared through DeepViFi², that does not use much domain-specific knowledge about viruses (Methods). Importantly, our model outperformed other methods in all measures, except for precision, for which DeepVirFinder outperformed all other methods (Figure 1b-c). However, precision is less critical for this framework because alignment steps are used to further filter out negatives. Importantly, DeepViFi², DeepVirFinder³, and ViraMiner⁴ were previously not trained or evaluated for RNA sequencing or 48bp reads, which is likely the reason that these methods are less appropriate in that context.”

Further, it should be acknowledged that not only bacteria infecting viral reads have been classified using deep learning, but also human infecting viral reads have been classified using machine learning models earlier, e.g HIVF (<https://doi.org/10.1111/tbed.13314>) or DeePac (<https://doi.org/10.1093/nargab/lqab004>).

We agree with the reviewer, and now refer to methods that classify human infecting viral reads in the revised Introduction section:

“More recently, methods have been developed to identify viruses that have potential to cause humans infections^{5,6}”

A2) influence of contaminants / false positive viral identifications

It can by no means be expected that the non-human and PhiX reads are all of viral origin, but many will be bacterial reads (and their phages), fungi etc. same as possible index swapping artefacts or run carryovers of Illumina machines. There are multiple approaches available to account for that on methodological level in the learning approaches, e.g., deepViFi has an open set approach, DeePac uses multiclass learning to distinguish them, but even more so quick a kraken (or similar) analysis of these reads to filter out non-likely viral reads appears

appropriate before actually running viRNAtrap to ensure that no reads of other origin contaminate the viral assembly and lead to potential false positive viral identifications.

We appreciate this comment. In this context, there at least two relevant meanings of 'contaminant'. We delineate two points addressing contamination:

1. One relevant meaning of 'contaminant' refers to contamination by many vectors of which phiX is but one example. For this purpose, we used vecscreen⁷ which does recognize numerous vectors that are partly of viral origin. We describe this in the revised Methods section:

"Filtering contaminants

To filter vector contaminants, we applied VeScreen⁷ to the assembled contigs that have been mapped to viruses through our databases, where virus accessions associated with vector contaminants were entirely removed from the search (Supplementary Dataset 11)."

We acknowledge that one can perform the vector contamination screen before the main steps of virRNAtrap instead of after, as in the current design. However, earlier vector contamination screening may lead to false negative vector contamination reads that could have been identified if assembled into longer sequences.

2. Another relevant meaning of 'contaminant' is sample contamination in which microbes that were not really in the cancer patient entered the sample while it was being processed in the laboratory. Unfortunately, it is not possible to model such contamination, as is also clearly mentioned in the deepViFi paper²:

"it is not possible currently to model all the contaminants"

We acknowledge this limitation in the revised Discussion section:

"However, it is not possible to model all contaminant of viruses that may have infected the samples during laboratory procedures²"

In addition to filtering such contaminants, it is implied that application of other software, such as Kraken, can filter out reads that are of non-viral origin. To evaluate this possibility, we applied Kraken2 to the LIHC (hepatocellular carcinoma) reads which were unmapped to human and the phiX phage in our pre-processing. We found that more than 83% of the reads were unclassified or mapped to 'unspecified taxonomy'. 13% of the reads were mapped to human, which are likely mutated and high complexity regions, that include the HERV sequences. 2% and 0.08% of the reads were mapped to root and cellular organisms, which include both viral and non-viral species. Therefore, more 99% of the reads would not be filtered with Kraken2 (Supplementary Figure 5). Furthermore, a quick search within the Kraken2 results has proven that many of the reads are incorrectly mapped to viral origin, for example, 17% of the samples have reads that are incorrectly mapped to the taxon *Alphabaculovirus*, a mapping that is not verified when using blastn. Therefore, we conclude that Kraken2 is not appropriate for filtering short reads of non-likely viral origin.

It is possible that the read length of 48bp is a problem for application of Kraken2, which was evaluated with a minimum read length of 100bp⁸. Indeed, longer sequences are known to be more accurately mapped⁹. Therefore, our current framework, which maps the assembled contigs, is likely to be more accurate in removing non-viral sequences.

We describe the Kraken2 computational experiment in the revised Methods section:

"In addition, we examined the application of software such as Kraken2⁸ to the RNAseq reads for filtering reads that are not likely of viral origin, by applying Kraken2 to reads of LIHC samples. However, we found that 99% of the reads would not be filtered using this approach (Supplementary Figure 5), likely due to the short reads (48bp) for which Kraken has not been designed or evaluated, as longer sequences are known to be more accurately mapped⁹."

Further, I could not find sufficient information on the quality standards for the viral assembly and blast analysis. What was the minimal contig length considered? How homogeneous was the coverage on the contigs? What were the standards of the blast analysis? Why were non-reference viruses only deeper investigated when occurring in more than one sample? It would

be very helpful if the authors provided quality information on the assembly for each identified virus.

We thank the reviewer for this comment. We comprehensively describe the parameters used for these analyses in the revised Methods section. The filtering we applied to minimize potential false positives is now described in detail. Of note, however, with this strict filtering, short contigs of divergent viruses may not be captured using Blastn. Therefore, as we now explain, non-reference viruses identified in multiple samples were further searched using STAR aligner because we reason that these are less likely to be contaminant or isolated events, but samples with fewer reads may be dismissed due to strict filtering.

The revised Methods section describing this now reads:

“Quality standards for virus identification

For all viruses, blastn was applied with E-value cutoff of 0.01 and any sequences with a match to contaminant accessions (that were associated with vector contamination) were filtered out.

- a. Reference viruses. For every sample, contigs mapped to each accession were extracted. Identified accessions with maximum qcov across contigs more than 90%, average qcov more than 50%, and average similarity more than 90% were considered. Accessions with maximal contig length under 100bp were manually inspected and verified against nr.
- b. Human endogenous viruses. For every sample, contigs mapped to each HERV were extracted. HERVs with contigs longer than 200bp, and with average qcov and similarity more than 95% were considered.
- c. Divergent viruses. For every sample, contigs mapped to each accession were extracted. Viruses already identified through the reference database were removed. Identified accessions with maximal contig length more than 300bp and qcov more than 40%, or with maximal contig length more than 100bp and qcov more than 75% and average similarity more than 75% were considered for manual inspection.

All instances of divergent viruses identified in TCGA samples were verified using blastn against nr, to support that the virus strain is indeed the best match to a viral contig generated by viRNAtrap. We reason that non-reference viruses (divergent viruses and viruses of non-human hosts) that were identified and verified in more than one sample were less likely to be contaminant or isolated events, whereas sample with fewer reads from such viruses may be filtered due to the strict filtering. We therefore additionally searched using the STAR aligner¹⁰ across tumor types where these viruses were identified through viRNAtrap (Supplementary Dataset 3). The following accessions were additionally searched using STAR to increase sample coverage (as these were the most interesting divergent strains found across multiple samples): Bermuda grass latent virus (NC_032405), *Armadillidium vulgare* iridescent virus IIV31 (NC_024451), *Geobacillus* virus (NC_009552) and the Human lung-associated videntovirus (NC_055523) “

A3) The analysis of the clinical outcome is interesting, but not entirely novel, e.g. <https://doi.org/10.3390/v12090956> studied HERV expression in tcga and their impact on survival. Surprisingly, this manuscript is cited with regard to the ancestry of HERV, but not in the context of survival (while it basically addresses the same question as the authors do, the authors claim that the impact of HERVs on cancer progression and clinical outcomes is not well understood). It would be helpful if the authors clarified these similarities

We agree with the reviewer. Thanks for pointing this out. In the revised manuscript, we clarify that HERV expression has been associated with poor survival in previous studies, and refer to several studies that reported such an association when discussing HERV in the Results section:

“Moreover, recent findings reported association between HERV expression and poor survival rates¹¹⁻¹⁵”

And:

“In agreement with previous studies¹¹⁻¹⁵, we find that patients with HERV-K- and HERV-H-positive cancer samples have significantly lower overall survival compared to HERV-K- and HERV-H-negative patients in COAD, LUSC, LUAD and LIHC”.

Minor:

B1) The software is not available as claimed by the authors in the Code and Software Submission Checklist. There was no demo on the authors' data available and scripts for reproduction which are claimed to be available in the submission checklist, are not given, rather the manuscript states they will be made available after publication (why not for review now?). Typical installation or run times are not given as stated.

We appreciate this comment. As part of this revision we:

1. We clearly provide a demo to test the software after installation, that is described in the main README.md of the GitHub page, as copied below:

Running example (Demo):

a. To run with an example input fastq file (`input_fastq/example_unmapped.fastq`) run

```
virnatrap-predict --input input_fastq/ --output output_contigs/
```

And evaluate the output file generated in `output_contigs/` using the expected output in `expected_output/output_py.txt`

There is a one-to-one correspondence between input files in directory `input_fastq/` and output files in directory `output_contigs` (or whatever subdirectories the user specifies). If an input file leads to zero predicted viral contigs, then the corresponding output file will be created but will be empty. The output files are in FASTA format but have the suffix `.txt` because experience has shown that Mac user prefer the suffix `.txt`. If one wants to rerun the command with the same input files and the same `output_contigs/` output directory, one should first remove the previous output files. `virnatrap-predict` will not overwrite output files that already exist.

The package comes with a small example that is intended to be used to test if one has installed `virNAttrap` correctly. The expected output is in subdirectory `expected_output`. To test if the above command worked as expected, run the additional command

```
diff expected_output/output_py.txt output_contigs/example_contigs.txt
```

to compare the output in the new installation to the expected output. The installation is correct if

2. We now provide all pre and post processing scripts, with clear instructions for how to download and process the data at every step, provided that dbGaP permission is granted. This is described in (<https://github.com/AuslanderLab/virnatrap/tree/main/scripts>) as copied below:

1. Pre-processing

A Requirements

(a) bowtie2: <https://github.com/BenLangmead/bowtie2>

(b) bam2fastq: <https://github.com/jts/bam2fastq>

(c) gdc-client: <https://github.com/topics/gdc-client>

B Instructions We provide an example manifest for download with 10 LIHC samples (manifest_LIHC_example.txt), where the token file path is defined as 'path-to-token'

B.1 To obtain the TCGA BAM files given dbGAP access, please follow the GDC instruction using gdc-client:

```
gdc-client download -m manifest_LIHC_example.txt -t path-to-token
```

B.2 Use bowtie to align against hg19 and the phix phage and get the unmapped reads. For that use the align_extract_reads.sh script . Set the paths to bowtie2 and bam2fastq, and change the following 3 lines in the align_extract_reads.sh script to define: (a) workdir - the working directory, where bam files are located. (b) bowtie2Indexhg19 - path to the hg19 index, and (c) bowtie2Indexphage - path to the phix index.

Then, running the align_extract_reads.sh script, it will create in workdir a directory called Results, with the paired-end files

B.2 Use the cat_paired.py scripts to copy paired-end files into one fastq file of all reads

2. Post-processing

B2) Similarly, it needs to be clarified how viral databases will be accessible (without restrictions) as they are currently not provided.

The viral databases are made available without restriction through the GitHub repository:

<https://github.com/AuslanderLab/virnatrap/tree/main/databases>

with a README.MD file documenting each of these databases, copied below.

..			
	README.MD	Update README.MD	24 days ago
	filtered_accessions.csv	Add files via upload	24 days ago
	herv_db.fa.txt	Add files via upload	24 days ago
	human_virus_db.fa	Add files via upload	24 days ago

README.MD 	
Viral databases Three databases that were used to search for viruses are provided (1) RefSeq reference human viruses, downloaded from the National Center for Biotechnology Information (NCBI) 76, to which we added human papillomaviruses strains that are not in RefSeq from PAVE (https://pave.niaid.nih.gov) - provided as human_virus_db.fa (2) more divergent viruses obtained from RVDB80 (https://hive.biochemistry.gwu.edu/rvdb/) which was then filtered to remove non-viral elements, endogenous viruses, and accessions that were consistently not verified using blastn against the nonredundant (nr) blast nucleotide database. The complete database is available for download C-RVDBv16.0.fasta from:https://hive.biochemistry.gwu.edu/prd/rvdb/content/C-RVDBv16.0.fasta Where accessions that were filtered out provided as filtered_accessions.csv (3) Human endogenous viruses. HERVd from (https://herv.img.cas.cz) where we found at least one identified retroviral protein motif of: POL/RT, GAG or ENV are provided as herv_db.fa	

B3) I could not find the word document specifying the primer design in the submission

Thank you. We now provide the primer design as part of the supplementary text. In addition, we added a section “Cells and culture conditions” to the Methods section, which now reads:

“Cells and culture conditions

Human ovarian cancer cell lines COV318 and OVI5E were cultured in RPMI1640 medium containing 10% fetal bovine serum (FBS) and 1% penicillin-streptomycin under 5% CO₂. All of the cell lines were authenticated at The Wistar Institute’s Genomics Facility using short-tandem-repeat DNA profiling. Regular mycoplasma testing was performed using a LookOut mycoplasma PCR detection kit (Sigma, cat. no. MP0035).

Experimental validation of the *Geobacillus* virus E2 in ovarian cancer cell lines.

Reverse-transcriptase qPCR (RT-qPCR) RNA was extracted using TRIzol reagent (Invitrogen, cat. no. 15596026). Extracted RNA was used for reverse-transcriptase PCR using a High-capacity cDNA reverse transcription kit (Thermo Fisher, cat. no. 4368814). Quantitative PCR was performed using a QuantStudio 3 real-time PCR system. GAPDH was used as an internal control. The fold change was calculated using the $2^{-\Delta\Delta C_t}$ method. The primers used for reverse-transcriptase qPCR are: GAPDH forward, GTCTCCTCTGACTTCAACAGCG and reverse, ACCACCCTGTTGCTGTAGTAGCCAA; *Geobacillus* virus E2 terminase forward, TTGCGATGCGTACTCAGACT and reverse, CTCTTTTTGGTCAGCAGCGG Primers were obtained using NCBI primer design tool as shown in the attached word document. The primers were synthesized by Integrated DNA Technologies IDT. The document specifying the primer design is provided in the Supplementary Information.

B4) Multiple Testing. The statistical methods section does not state that any multiple testing correction was performed. Although the manuscript (e.g. in figure 2) only shows a limited number of tests, these include some rare choices of viruses (e.g. torquevirus) making me assume that the authors tested all viruses on all cancer types (or later on all HERVs etc.). Most p-values shown in figure 2 for instance are not significant any longer after Bonferroni correction for the roughly 50 observed (and 140 theoretically possible) combinations of cancer tissues and viral families. There are smarter ways to do multiplicity correction than Bonferroni and maybe the authors have some (unstated) assumptions to reduce the hypothesis space, but with the current information provided these findings should not be considered statistically significant. Further, there should be a full reporting of all results (e.g. in the supplement) not only those chosen for discussion.

We agree with the reviewer and now revise this analysis to correct for multiple testing. We indeed focus on a subset comparison with at least 5 positive cases and focus on known oncoviruses for the reference viruses survival analysis. We perform correction within each cancer type (which retains the known association between HPV and improved HNSC survival,

as well as seven HERVs that are significantly associate with survival), and a global correction across all cancer types (which retains three HERVs that are significantly associated with survival). We additionally provide the results for all tested associations in Supplementary Table 1 and in Supplementary Dataset 12. The revised Methods section reads:

“Viruses with significant log-rank p-values are reported as significantly associated with survival. We tested associations for each cancer type and evaluated those with at least 5 cases in each group. We applied FDR correction within each cancer type and additionally applied a global FDR correction for all comparisons across cancer types.

For reference viruses in Figure 2, we focused on known oncoviruses, HR-HPV, HCV, and HBV, thus testing at most one hypothesis for each cancer type except for LIHC where two hypotheses were tested (Supplementary Table 1 and Supplementary Figure 2). Therefore, none of the reference viruses were significantly associated with survival after global FDR correction, whereas only HR-HPV was significant for HNSC specific correction.

HERVs that were identified in least 5 TCGA samples (Supplementary Dataset 4) we correlated with survival (Figure 3 and Supplementary Dataset 12), and p-values were corrected in a cancer-type specific manner (yielding seven significant associations) and globally across all comparison (yielding three significant associations, Figure 3).“

Reviewer #2 (Remarks to the Author):

The authors present a novel pipeline to analyse viral reads in human RNA sequencing data sets. Based on a two step approach they first use a deep learning model trained on human and viral reads to identify potential viral reads. In a second step these reads are assembled using a simple overlap based approach. The resulting contigs are then analysed with blastn to identify viral sequences.

Based on the data presented it is difficult to assess the performance of their approach as a description of the read support for the individual viral detections is missing.

Major

- The authors seem to only present virus positivity (1) or negativity (0) in the supplementary tables 3,4 and 9. For an evaluation of virus positivity the number of supporting reads would be very helpful.

We thank the reviewer for this comment. In the revised manuscript we:

(1) Provide a dataset with the complete information and statistics for identified viruses reported in the original supplementary tables 3,4 and 9, in Supplementary Dataset 3. This dataset provides the accession of the virus, full name of the virus, the average blastn similarity score and coverage for contigs mapped to the virus, and a list with contigs lengths. Because our method assembles reads into contigs and removes reads that are overlapping with contigs that have been assembled, it is difficult to quantify the number of reads through viRNAtrap. However, we provide the number of reads for viruses that have been verified using STAR in Supplementary Dataset 3 as well.

(2) comprehensively describe the parameters used for these analyses and the filtering we applied to remove contaminants and to minimize potential false positive.

The revised Methods section describing this now reads:

"Quality standards for virus identification"

For all viruses, blastn was applied with E-value cutoff of 0.01 and any sequences with a match to contaminant accessions (that were associated with vector contamination) were filtered out.

- a. Reference viruses. For every sample, contigs mapped to each accession were extracted. Identified accessions with maximum qcov across contigs more than 90%, average qcov more than 50%, and average similarity more than 90% were considered. Accessions with maximal contig length under 100bp were manually inspected and verified against nr.

- b. Human endogenous viruses. For every sample, contigs mapped to each HERV were extracted. HERVs with contigs longer than 200bp, and with average qcov and similarity more than 95% were considered.
- c. Divergent viruses. For every sample, contigs mapped to each accession were extracted. Viruses already identified through the reference database were removed. Identified accessions with maximal contig length more than 300bp and qcov more than 40%, or with maximal contig length more than 100bp and qcov more than 75% and average similarity more than 75% were considered for manual inspection.

All instances of divergent viruses identified in TCGA samples were verified using blastn against nr, to support that the virus strain is indeed the best match to a viral contig generated by viRNAtrap. We reason that non-reference viruses (divergent viruses and viruses of non-human hosts) that were identified and verified in more than one sample were less likely to be contaminant or isolated events, whereas sample with fewer reads from such viruses may be filtered due to the strict filtering. We therefore additionally searched using the STAR aligner¹⁰ across tumor types where these viruses were identified through viRNAtrap (Supplementary Dataset 3). The following accessions were additionally searched using STAR to increase sample coverage (as these were the most interesting divergent strains found across multiple samples): Bermuda grass latent virus (NC_032405), *Armadillidium vulgare* iridescent virus IIV31 (NC_024451), *Geobacillus* virus (NC_009552) and the Human lung-associated vientovirus (NC_055523).“

- The authors could compare their approach to a simple assembly of the extracted reads and analysis of the resulting contigs. This would enable to evaluate whether they lose virus supporting reads and thereby sensitivity by their approach and enable to evaluate the speed improvement achieved.

We thank the reviewer for this important comment. In the revised manuscript, we perform such comparison analysis for 10 LIHC samples (Supplementary Figure 4 and

Supplementary Table 2). We find that while the running time of the naïve assemble is substantially longer (up to 6 times longer, supplementary Figure 4), the same viruses are identified, albeit with different number of detected contigs (the model-based approach led to fewer contigs identified).

To further evaluate the sensitivity of the pipeline based on the number of viral reads that are present in a sample, we additionally perform a simulated analysis. We empirically evaluate the number viral reads required for identification with the current seed threshold of 0.7, and find that in 93% of the cases, more than 5 reads are sufficient, and in 99% of the cases, 10 reads are sufficient (Supplementary Figure 4).

These analyses are now described in the revised Methods section:

“viRNAtrap performance evaluation

To evaluate the contribution of the model to the viRNAtrap pipeline we re-ran viRNAtrap on 10 LIHC samples, and additionally ran a modified viRNAtrap pipeline not using the model, on the same system. We compared the viruses identified by the model-based approach to those that are identified when the pipeline is applied without using the model (Supplementary Table 2, showing similar viruses with different number of contigs). We additionally compared the running time of the two approaches (Supplementary Figure 4).

To evaluate the sensitivity of the viRNAtrap pipeline based the number of viral reads present in a sample, we performed a simulated analysis. From the test dataset, we down sampled groups of viral reads with different group sizes (10,000 groups for each size, from one read up to 10 reads), and we evaluated the number of groups with at least one read that is scored above 0.7, which is the seed threshold used for the viRNAtrap assembly. Therefore, this analysis is estimating the probability of identifying viruses based on the number of reads present. We found 93% and 99% of the groups with more than 5 and 9 reads, respectively, would be identified.”

And these results are mentioned in the revised Discussion section:

“We further show that using the deep learning model substantially improves the running time, while not compromising sensitivity if more than 5 viral reads are present (Supplementary Figure 4, see Methods).”

- The authors restrict their analysis to the reads not mapped to the human reference genome. It is unclear to me how this affects the detection of HERVs that are present in the human reference genome and should therefore not be detected in an optimal way.

Thank you for this comment. HERV are rarely captured by alignment to the human genome, because of their high complexity and high mutation rate (Bannert N et al, PNAS 2004, PMID: 15310846, which is reference 17 below). Therefore, alignment with 2 allowed mismatches per read performed in the pre-processing step, would not capture these genomic regions, that are usually handled independently^{15,16}. We now explain this, and acknowledge that in rare cases, unmutated HERV sequences will not be identified by this pipeline.

“Importantly, high mutation rate of HERV¹⁷ prohibits most HERV sequences from aligning to the human genome in pre-processing^{15,16}, however, in rare cases, HERV regions that are conserved would not be identified by this approach.”

Minor

- Did the authors correct for multiple testing throughout the manuscript when testing for associations e.g. across entities or mutated genes

Thank you. For associations with mutated genes (through Figure 3 and Supplementary Dataset 6) we only report negative results (insignificant associations) because no significant associations were identified after correction for multiple hypothesis testing. We now additionally revise the correction for multiple hypotheses for survival analyses. We perform correction within each cancer type (which retains the known association between HPV and better HNSC survival, as well as seven HERVs that are significantly associated with survival), and a global correction across all cancer types (which retains three HERVs that are significantly associated with survival). The revised Methods section reads:

“Viruses with significant log-rank p-values are reported as significantly associated with survival. We tested associations for each cancer type and evaluated those with at least 5 cases in each group. We applied FDR correction within each cancer type and additionally applied a global FDR correction for all comparisons across cancer types.

For reference viruses in Figure 2, we focused on known oncoviruses, HR-HPV, HCV, and HBV, thus testing at most one hypothesis for each cancer type except for LIHC where two hypotheses were tested (Supplementary Table 1 and Supplementary Figure 2). Therefore, none of the reference viruses were significantly associated with survival after global FDR correction, whereas only HR-HPV was significant for HNSC specific correction.

HERVs that were identified in least 5 TCGA samples (Supplementary Dataset 4) we correlated with survival (Figure 3 and Supplementary Dataset 12), and p-values were corrected in a cancer-type specific manner (yielding seven significant associations) and globally across all comparison (yielding three significant associations, Figure 3).“

References

- 1 Keras (2015).
- 2 Rajkumar, U. *et al.* in *Proceedings of the 13th ACM International Conference on Bioinformatics, Computational Biology and Health Informatics* 1-8 (2022).
- 3 Ren, J. *et al.* Identifying viruses from metagenomic data using deep learning. *Quant Biol* **8**, 64-77, doi:10.1007/s40484-019-0187-4 (2020).
- 4 Tampuu, A., Bzhalava, Z., Dillner, J. & Vicente, R. ViraMiner: Deep learning on raw DNA sequences for identifying viral genomes in human samples. *PLoS One* **14**, e0222271, doi:10.1371/journal.pone.0222271 (2019).
- 5 Zhang, Z. *et al.* Rapid identification of human-infecting viruses. *Transbound Emerg Dis* **66**, 2517-2522, doi:10.1111/tbed.13314 (2019).
- 6 Bartoszewicz, J. M., Seidel, A. & Renard, B. Y. Interpretable detection of novel human viruses from genome sequencing data. *NAR Genom Bioinform* **3**, lqab004, doi:10.1093/nargab/lqab004 (2021).

- 7 Schäffer, A. A. *et al.* VecScreen_plus_taxonomy: imposing a tax(onomy) increase on vector contamination screening. *Bioinformatics* **34**, 755-759, doi:10.1093/bioinformatics/btx669 (2018).
- 8 Wood, D. E., Lu, J. & Langmead, B. Improved metagenomic analysis with Kraken 2. *Genome Biol* **20**, 257, doi:10.1186/s13059-019-1891-0 (2019).
- 9 Celaj, A., Markle, J., Danska, J. & Parkinson, J. Comparison of assembly algorithms for improving rate of metatranscriptomic functional annotation. *Microbiome* **2**, 39, doi:10.1186/2049-2618-2-39 (2014).
- 10 Dobin, A. *et al.* STAR: ultrafast universal RNA-seq aligner. *Bioinformatics* **29**, 15-21, doi:10.1093/bioinformatics/bts635 (2013).
- 11 Hahn, S. *et al.* Serological response to human endogenous retrovirus K in melanoma patients correlates with survival probability. *AIDS Res Hum Retroviruses* **24**, 717-723, doi:10.1089/aid.2007.0286 (2008).
- 12 Kolbe, A. R. *et al.* Human Endogenous Retrovirus Expression Is Associated with Head and Neck Cancer and Differential Survival. *Viruses* **12**, doi:10.3390/v12090956 (2020).
- 13 Zhao, J. *et al.* Expression of Human Endogenous Retrovirus Type K Envelope Protein is a Novel Candidate Prognostic Marker for Human Breast Cancer. *Genes Cancer* **2**, 914-922, doi:10.1177/1947601911431841 (2011).
- 14 Reis, B. S. *et al.* Prostate cancer progression correlates with increased humoral immune response to a human endogenous retrovirus GAG protein. *Clin Cancer Res* **19**, 6112-6125, doi:10.1158/1078-0432.Ccr-12-3580 (2013).
- 15 Zapatka, M. *et al.* The landscape of viral associations in human cancers. *Nat Genet* **52**, 320-330, doi:10.1038/s41588-019-0558-9 (2020).
- 16 Smith, C. C. *et al.* Endogenous retroviral signatures predict immunotherapy response in clear cell renal cell carcinoma. *J Clin Invest* **128**, 4804-4820, doi:10.1172/jci121476 (2018).
- 17 Bannert, N. & Kurth, R. Retroelements and the human genome: new perspectives on an old relation. *Proc Natl Acad Sci U S A* **101 Suppl 2**, 14572-14579, doi:10.1073/pnas.0404838101 (2004).

REVIEWER COMMENTS

Reviewer #1 (Remarks to the Author):

Overall, the authors have greatly improved the manuscript. Most of my points have been addressed, I have a minor question still regarding (A1), however, I do have significant worries regarding the answer to my minor aspect (B4) regarding multiplicity adjustment. Maybe I am missing something here (there were too few details reported in the original submission and still are too few details reported now):

A1)

The comparison with competing methods is very insightful. It lacks however details, in particular how hyperparameter tuning was performed for the other methods and which parameters were selected in the end. It would be important to make this accessible to show that a fair comparison was conducted.

B4)

I have difficulties following and approving the statistical analysis. From my understanding, this may have a flavor of Texas barn shooting. The authors conducted an analysis in the originally submitted manuscript. They did not do any multiplicity correction. Now, posthoc they define exclusion criteria (which appear to contradict the original analysis performed) and redo the analysis, now finding significance which would not have been possible in the original design. The claim of statistical significance may not valid in this study design and may not be trustworthy.

Further, why is the lowest p-value candidate in Fig 3b) and 4 in the current analysis not reported in the original analysis?

The legend for 3b and 4 states that "The log rank and proportional hazards (PH) p-values are reported", I can however only find one p-value in the plot. Which one is reported and why is the second one missing? Again, is there a reason in the study design not to report them any longer or may this be due to the fact that they are no longer significant? This again, would be cherry-picking.

Why are results that have highly significant p-values now in figure 3b not appearing in the original submission (if testing standards are stricter now so that they are significant – why were they not significant before)?

Overall, the reporting of the statistical procedures need to be more in detail and should be restricted to those cases where there were clear pre-analysis hypotheses to be tested. Ideally, the authors

share their original statistical study design and a point-by-point explanation of changes made (and their respective statistical justification).

Reviewer #2 (Remarks to the Author):

The authors now included a comparison to other tools trained to identify viruses (DeepViFi, DeepVirFinder, ViraMiner and 'off-the-shelf Seq2Seq'. The supplementary information now includes a more detailed description of the blastn hits. They in addition improved the correction for multiple testing by applying FDR. Furthermore the documentation of the software tool on Github is substantially improved.

We thank both reviewers for the additional comments on our revised manuscript NCOMMS-22-33903A. We now address these comments in the revised manuscript and better describe the statistical analysis. All text changes related to any reviewer comment are in red font.

Reviewer #1 (Remarks to the Author):

Overall, the authors have greatly improved the manuscript. Most of my points have been addressed, I have a minor question still regarding (A1), however, I do have significant worries regarding the answer to my minor aspect (B4) regarding multiplicity adjustment. Maybe I am missing something here (there were too few details reported in the original submission and still are too few details reported now):

A1)

The comparison with competing methods is very insightful. It lacks however details, in particular how hyperparameter tuning was performed for the other methods and which parameters were selected in the end. It would be important to make this accessible to show that a fair comparison was conducted.

We appreciate this comment. We now provide the complete information describing how other methods were implemented in the supplementary, which reads:

"Training existing methods for virus identification"

1. *DeepViFi*. We trained DeepViFi as instructed in the method's github repository, <https://github.com/UCRajkumar/DeepViFi>. A transformer was trained using the parameters defined in the configuration file, with embedding dimension of 128, 16 heads, 8 layers, the feed forward dimension set to 256 and the batch size set to 256. The generated embedding by the transformer for each sequence read was used to train a

random forest classifier using the transformer representation (through sklearn.ensemble), with 500 trees as recommended by DeepViFi.

2. *DeepVirFinder*. We followed the instructions of DeepVirFinder github repository:

<https://github.com/jessieren/DeepVirFinder> to train a model and evaluate it using our data. Even though DeepVirFinder was developed to take various input sizes (300bps, 500bps and 1000bps), there is an option to choose input size less than 300bps, which we used by setting the input size to 48. We used the parameters as defined by the authors to train the model as following: dropout convolutional neural network (CNN) of 0.1, dropout pool of 0.1, learning rate of 0.001 and number of filters of 500, of which each of length of 10.

3. *ViraMiner*. The ViraMiner model was trained as end-to-end CNN model as instructed in its github repository, <https://github.com/NeuroCSUT/ViraMiner>. The model was trained with filter size 8, dropout of 0.1, learning rate of 0.001 and layer_size of 1000. Even though the input sequence length in the original method was defined to be 300bps, we modified the code (specifically, we modified helper_with_n.py line 73 from 300 to 48) to accept input sequences of size 48bps.

4. *Off-the-shelf seq2seq*. We trained off-the-shelf seq2seq model using Keras (with LSTM components) on our data by configuring the model to take 48bp input sequences and the embedding size was defined to be of size 64 while the learning rate was set to 0.001. Then, to accommodate to DeepViFi, which also compared their representation against off-the-shelf seq2seq model, the seq2seq representation of viral sequences was given as input to a random forest classifier (using sklearn.ensemble) with the same parameter defined, the number of trees, to be 500.“

B4)

I have difficulties following and approving the statistical analysis. From my understanding, this may have a flavor of Texas barn shooting. The authors conducted an analysis in the originally submitted manuscript. They did not do any multiplicity correction. Now, posthoc they define exclusion criteria (which appear to contradict the

original analysis performed) and redo the analysis, now finding significance which would not have been possible in the original design. The claim of statistical significance may not valid in this study design and may not be trustworthy.

We thank the referee for this multi-part comment. We split the comment into its parts and interleaved our responses.

We understand why the description of the revised analysis raised valid concerns.

Reviewer 1 is correct in that in the original submission we reported the survival analysis p-values without correction for multiple hypotheses, but the reviewer comments should be about the revision.

In the revised analysis, the exclusion criterion was not applied to increase the number of significant associations, in fact, we would be getting many more significant associations if not applying the criterion requiring 5 cases. When revising the analysis in our first revision to correct for multiple testing, we noticed the issue that many of the highly significant association had 1-3 positive samples, and therefore decided to apply the exclusion criterion. As the reviewer suggests, we now describe and provide both analyses (with and without the exclusion criterion) in Supplementary Dataset 12.

Further, why is the lowest p-value candidate in Fig 3b) and 4 in the current analysis not reported in the original analysis?

In the original analysis of Figure 3, where we did not apply correction for multiple hypotheses, many significant associations were identified (all with poor survival). We therefore selected a subset of HERV to present, not by the p-value (which seemed arbitrary), but some of those that appeared in cancer types where we found multiple significant (non-corrected) associations. In fact, all four LIHC association were also presented in the original figure 3. In the revised analysis, when applying the exclusion criteria and the FDR correction, we find many fewer significant associations that are reported in in the revised Figure 3.

Figure 4 panels are not changed from the original submission because correction for testing multiple hypotheses is not needed in Figure 4 where the examined associations are all for IIV31 presence.

The legend for 3b and 4 states that “The log rank and proportional hazards (PH) p-values are reported”, I can however only find one p-value in the plot. Which one is reported and why is the second one missing? Again, is there a reason in the study design not to report them any longer or may this be due to the fact that they are no longer significant? This again, would be cherry-picking.

In the revised analysis we realized that using PH is less exact for some of the examined associations, specifically less fitting for comparisons with small sample sizes. The log-rank is reported since it is nonparametric and more exact in small sample cases, which are a substantial subset of the cases evaluated. Given this, and since PH and log rank are approximately the same asymptotically, and thus redundant, (e.g., when sample size is large enough), we remove the PH p-value evaluation and report the log-rank gold standard for all survival analyses. The remaining mention of PH in the legends of our first revision was therefore a syntax error that has been corrected. We thank the reviewer for catching this error.

Why are results that have highly significant p-values now in figure 3b not appearing in the original submission (if testing standards are stricter now so that they are significant – why were they not significant before)?

As explained above, In the original analysis of Figure 3, where we did not apply correction for multiple hypotheses, many significant associations were identified, all with poor survival. We therefore selected a subset of HERV to present, not by the p-value, but we presented a subset of those that appear in cancer types where we found multiple significant (non-corrected) associations. In the revised analysis, when applying the exclusion criteria and the FDR correction, we find many fewer significant associations which are reported in Figure 3. Overall, the reviewer is correct in that the significant

associations in the current figure 3 were also significant when not applying FDR correction.

Overall, the reporting of the statistical procedures need to be more in detail and should be restricted to those cases where there were clear pre-analysis hypotheses to be tested. Ideally, the authors share their original statistical study design and a point-by-point explanation of changes made (and their respective statistical justification).

We appreciate and agree with this comment. In the second revision of the manuscript, we better describe the process leading to this analysis in the Methods section:

“Viruses with significant log-rank p-values are reported as significantly associated with survival. Our examination led to multiple significant associations between survival and viral presence for cases where 1-3 virus-positive samples were found. We therefore revised our analysis and tested associations for each cancer type, evaluating those with at least 5 cases in each group; the decision to use 5 as the lower bound rather than 4 was made because 5 is considered a ‘round number’. We applied FDR correction within each cancer type and additionally applied a global FDR correction for all comparisons across cancer types.

For reference viruses in Figure 2, none of the reference viruses were significantly associated with survival after global FDR correction, whereas only HR-HPV was significant for HNSC specific correction. While this significance is mild, we report this association because it is confirmatory of a known association between HR-HPV and HNSC survival^{33,34}.

HERVs that were identified in at least 5 TCGA samples (Supplementary Dataset 4) were correlated with survival (Figure 3 and Supplementary Dataset 12), and p-values were corrected in a cancer-type specific manner (yielding seven significant associations) and globally across all comparison (yielding three significant associations, Figure 3). Importantly, our examination revealed multiple significant HERV-survival associations for cases with 1-4 HERV positive samples, which could be of interest, but did not seem sufficiently reliable, therefore leading us to

set the five sample cutoff. For completeness, we additionally report the FDR corrected p-values without applying this restriction in Supplementary Dataset 12.”

Reviewer #2 (Remarks to the Author):

The authors now included a comparison to other tools trained to identify viruses (DeepViFi, DeepVirFinder, ViraMiner and 'off-the-shelf Seq2Seq'. The supplementary information now includes a more detailed description of the blastn hits. They in addition improved the correction for multiple testing by applying FDR. Furthermore the documentation of the software tool on Github is substantially improved.

We thank reviewer 2 for the first round comments and suggestions that helped us to greatly improve our manuscript.

Reviewers' comments:

Reviewer #1 (Remarks to the Author):

The authors have provided a detailed response to the two issues I raised. While I know better understand their reasoning, I do not necessarily agree with the conclusions drawn.

Benchmarking:

The authors now describe the training of competing tools. However, it becomes clear that even though they use tools on settings that they were not trained for (e.g. DeepVirFinder on 48bp reads) they did not do any hyperparameter optimization, but relied on default parameters. It is very likely that results for other tools will increase massively.

Multiple Testing

The authors need to decide what the goal of their study. Is this an exploratory data analysis or a hypothesis-driven significance test?

In the first scenario, the way the authors proceed is absolutely valid, to look at results, to readjust hypothesis and to add exclusion criteria after looking at results to focus on the most relevant ones. But if the authors need to do this, this needs to be clearly labeled as an exploratory data analysis (and I think it still would be interesting).

However, if the authors want to claim statistical significance for their findings in a hypothesis-driven setup, these hypotheses and filters cannot easily be adjusted post hoc if results had been significant before the filtering criteria were applied. Given the current reporting, this does not appear to be true for all cases. I have therefore doubts whether claims of statistical significance are correct.

We thank the reviewer for their additional comments on our revised manuscript, which we address in this revision.

Reviewer #1 (Remarks to the Author):

The authors have provided a detailed response to the two issues I raised. While I know better understand their reasoning, I do not necessarily agree with the conclusions drawn.

Benchmarking:

The authors now describe the training of competing tools. However, it becomes clear that even though they use tools on settings that they were not trained for (e.g. DeepVirFinder on 48bp reads) they did not do any hyperparameter optimization, but relied on default parameters. It is very likely that results for other tools will increase massively.

We respectfully point that the request of the referee to perform hyperparameter search for existing models is unreasonable.

The referee originally requested that we compare against other methods DeepViFi¹, DeepVirFinder², ViraMiner³, as well as 'off-the-shelf Seq2Seq' compared through DeepViFi¹. None of these methods was designed nor tested for RNAseq or reads shorter than 150bp. To avoid applying models to data they were not fit to process, we put substantial effort to retrain the models using our data, which is **going beyond the standard practice when comparing to previous approaches**, not done by DeepVirFinder², or multiple other studies for phage identification that never retrained a published model⁴⁻⁶. Further, even a recent benchmarking study of these approaches did not retrain any published model⁷.

A provided model is considered representative of a published method for respective comparisons not only for virus identification, but throughout areas of biomedicine, where comparison to previously published machine and deep learning models are using the published model without retraining⁸⁻¹⁴.

DeepViFi is the one study that retrained models it compared against, however, even DeepViFi did not perform hyperparameter search for published models. We quote from the DeepViFi publication: “We retrained DeepVirFinder and ViraMiner model on a custom training set before evaluation (Methods). Despite retraining, ViraMiner and DeepVirFinder both achieved an AUC value of less than 0.5 on all 4 test sets”.

Performing a new hyperparameter search with a different training data will lead to *entirely different model architectures*. It is possible that some of such new architectures would be similar to and have performances close to the best model we found through our comprehensive hyper-parameter optimization, but such new architectures are distinct from and not representative of the original published method.

Multiple Testing

The authors need to decide what the goal of their study. Is this an exploratory data analysis or a hypothesis-driven significance test?

In the first scenario, the way the authors proceed is absolutely valid, to look at results, to readjust hypothesis and to add exclusion criteria after looking at results to focus on the most relevant ones. But if the authors need to do this, this needs to be clearly labeled as an exploratory data analysis (and I think it still would be interesting).

We appreciate this comment. Indeed, this study provides a new approach and generally performs an exploratory data analysis using this approach. We now explicitly mention this in the revised manuscript, in the abstract:

“We utilize viRNAtrap, which is based on a deep learning model trained to discriminate viral RNAseq reads, to explore viral expression in cancers and apply it to 14 cancer types from The Cancer Genome Atlas (TCGA).

In the introduction:

“We apply viRNAtrap to 14 cancer types from TCGA (selected based on potential viral relevance to oncogenesis), to perform an exploratory data analysis and characterize the landscape of viral infections in the human cancer transcriptome.”

And in the discussion:

“We employ viRNAtrap for an exploratory data analysis and characterize viruses that are expressed across 14 cancer tissues from TCGA and analyze their genomic and survival correlates.”

For included hypothesis-driven analyses, we eliminate any exclusion criteria in the revised manuscript that were considered after the original submission, and consider the data completely. We report significant results only for unfiltered analyses that were originally considered, with correction for multiple testing. This is described in the next section with more detail.

However, if the authors want to claim statistical significance for their findings in a hypothesis-driven setup, these hypotheses and filters cannot easily be adjusted post hoc if results had been significant before the filtering criteria were applied. Given the current reporting, this does not appear to be true for all cases. I have therefore doubts whether claims of statistical significance are correct.

Our exploratory data analysis uncovered some significant associations even with the complete hypothesis testing and no filtering after correcting for testing multiple comparisons. While none of the survival associations originally in Figure 2 was significant without filtering, the seven associations provided in revised Figure 3 were

significant without the filtering. To resolve any confusion, in the revised manuscript we exclude any filtering criteria, and we report as significant only the associations that are significant with no filtering applied. Therefore, in the revised manuscript, all claims of statistical significance are correct and justified:

- 1) We clearly mention that none of the associations with reference oncoviruses were significant after correction for multiple hypotheses, and that the association between HNSC and HR-HPV is reported with unadjusted p-value because it is confirmatory of a known association. This is clarified in the results section:

“While none of the associations were significant after adjustment for multiple hypotheses (Supplementary Figure 2, Supplementary Table 1), we found that HR- α HPV-positive HNSC patients have better survival compared to HR- α HPV-negative patients (by the Kaplan Meier curves Figure 2c), which is confirmatory of previous studies^{15,16}.”

And in the methods section:

“None of the reference viruses were significantly associated with survival after FDR correction, however, we report in Figure 2 the association between HR-HPV with unadjusted p-value because it is confirmatory of a known association between HR-HPV and HNSC survival^{15,16}.”

- 2) We report significant associations between HERV and survival without any filtering, using two types of FDR-correction: once when correcting for each cancer type individually, and second when correcting for all cancer types together. All significant associations (FDR adjusted p-value < 0.05) are reported in Supplementary Dataset 12, and selected significant associations are displayed in Figure 3:

“For HERV analysis, we present in the main text selected associations with at least 5 cases in each group, where additional significant associations between survival and viral presence are reported in the Supplementary Dataset 12 “

References

- 1 Rajkumar, U. *et al.* in *Proceedings of the 13th ACM International Conference on Bioinformatics, Computational Biology and Health Informatics* 1-8 (2022).
- 2 Ren, J. *et al.* Identifying viruses from metagenomic data using deep learning. *Quant Biol* **8**, 64-77 (2020). <https://doi.org:10.1007/s40484-019-0187-4>
- 3 Tampuu, A., Bzhalava, Z., Dillner, J. & Vicente, R. ViraMiner: Deep learning on raw DNA sequences for identifying viral genomes in human samples. *PLoS One* **14**, e0222271 (2019). <https://doi.org:10.1371/journal.pone.0222271>
- 4 Fang, Z. *et al.* PPR-Meta: a tool for identifying phages and plasmids from metagenomic fragments using deep learning. *Gigascience* **8** (2019). <https://doi.org:10.1093/gigascience/giz066>
- 5 Kieft, K., Zhou, Z. & Anantharaman, K. VIBRANT: automated recovery, annotation and curation of microbial viruses, and evaluation of viral community function from genomic sequences. *Microbiome* **8**, 90 (2020). <https://doi.org:10.1186/s40168-020-00867-0>
- 6 Bai, Z. *et al.* Identification of bacteriophage genome sequences with representation learning. *Bioinformatics* **38**, 4264-4270 (2022). <https://doi.org:10.1093/bioinformatics/btac509>
- 7 Ho, S. F. S., Wheeler, N., Millard, A. D. & van Schaik, W. Comprehensive benchmarking of tools to identify phages in metagenomic shotgun sequencing data. (2022). <https://doi.org:10.1101/2021.04.12.438782>
- 8 Quang, D. & Xie, X. DanQ: a hybrid convolutional and recurrent deep neural network for quantifying the function of DNA sequences. *Nucleic Acids Res* **44**, e107 (2016). <https://doi.org:10.1093/nar/gkw226>
- 9 Avsec, Ž. *et al.* Effective gene expression prediction from sequence by integrating long-range interactions. *Nat Methods* **18**, 1196-1203 (2021). <https://doi.org:10.1038/s41592-021-01252-x>
- 10 Zhou, J. *et al.* Deep learning sequence-based ab initio prediction of variant effects on expression and disease risk. *Nat Genet* **50**, 1171-1179 (2018). <https://doi.org:10.1038/s41588-018-0160-6>
- 11 Mortuza, S. M. *et al.* Improving fragment-based ab initio protein structure assembly using low-accuracy contact-map predictions. *Nat Commun* **12**, 5011 (2021). <https://doi.org:10.1038/s41467-021-25316-w>
- 12 Wang, D. *et al.* Optimized CRISPR guide RNA design for two high-fidelity Cas9 variants by deep learning. *Nat Commun* **10**, 4284 (2019). <https://doi.org:10.1038/s41467-019-12281-8>

- 13 Ahsan, M. U., Liu, Q., Fang, L. & Wang, K. NanoCaller for accurate detection of SNPs and indels in difficult-to-map regions from long-read sequencing by haplotype-aware deep neural networks. *Genome Biol* **22**, 261 (2021). <https://doi.org:10.1186/s13059-021-02472-2>
- 14 Dohmen, J. *et al.* Identifying tumor cells at the single-cell level using machine learning. *Genome Biol* **23**, 123 (2022). <https://doi.org:10.1186/s13059-022-02683-1>
- 15 Sethi, S. *et al.* Characteristics and survival of head and neck cancer by HPV status: a cancer registry-based study. *Int J Cancer* **131**, 1179-1186 (2012). <https://doi.org:10.1002/ijc.26500>
- 16 Sarkar, S. *et al.* Human papilloma virus (HPV) infection leads to the development of head and neck lesions but offers better prognosis in malignant Indian patients. *Med Microbiol Immunol* **206**, 267-276 (2017). <https://doi.org:10.1007/s00430-017-0502-5>